# A spatial long-read approach at near-single-cell resolution reveals developmental regulation of splicing and polyadenylation sites in distinct cortical layers and cell types

Careen Foord [1,2,11], Andrey D. Prjibelski[3,11], Wen Hu [1,2,11], Lieke Michielsen [1,2,4,5], Andrea Vandelli [6], Oleksandr Narykov[7,8,9], Brian Evans[1,2], Justine Hsu[1,2], Natan Belchikov[1,2,10], Julien Jarroux [1,2], Yi He[1,2], M. Elizabeth Ross [1,2], Iman Hajirasouliha [4,5], Gian Gaetano Tartaglia [6], Dmitry Korkin [7,8,9], Alexandru I. Tomescu [3] & Hagen U. Tilgner [1,2] ✉

Genome-wide spatial long-read approaches often lack single-cell resolution and yield limited read lengths. Here, we introduce spatial ISOform sequencing (Spl-ISO-Seq), which reveals exons and polyadenylation sites with near-single-cell resolution. Spl-ISO-Seq selects long cDNAs and doubles to triples read lengths compared to standard preparations. Adding a highly specific software tool (Spl-ISOquant) and comparing human post-mortem pre-puberty (8–11 years) to post-puberty (16–19 years) visual cortex samples, we find that cortex harbors stronger splicing and poly(A)-site regulation than white matter. However, oligodendrocyte regulation is stronger in white matter. Among cortical layers, layer 4 has the most developmentally-regulated splicing changes in excitatory neurons and in poly(A) sites. We also find repeat elements downstream of developmentally-regulated layer 4 exons. Overall, alternative splicing changes are linked to post-synaptic structure and function. These results root developmental splicing changes during puberty in specific layers and cell types. More generally, our technologies enable exciting observations for any complex tissue.

Alternative splicing (AS) affects nearly all human protein-coding genes[1-4], impacting neuronal properties, growth, excitability, synapse specification, plasticity, and neuronal network function[5-8]. Moreover, AS is associated with aging and developmental brain diseases[6,7,9-14].

Likewise, polyadenylation (poly(A))-site changes are abundant genome-wide and have been linked to cellular function[15-19]. The naming of long-read sequencing as the method of the year in January of 2023[20-25] highlights this technology's ability to record such RNA

[1]Feil Family Brain and Mind Research Institute, Weill Cornell Medicine, New York, NY, USA. [2]Center for Neurogenetics, Weill Cornell Medicine, New York, NY, USA. [3]Department of Computer Science, University of Helsinki, Helsinki, Finland. [4]Institute for Computational Biomedicine, Department of Physiology and Biophysics, Weill Cornell Medicine of Cornell University, New York, NY, USA. [5]Caryl and Israel Englander Institute for Precision Medicine, The Meyer Cancer Center, Weill Cornell Medicine, New York, NY, USA. [6]RNA Systems Biology Lab, Center for Human Technologies, Istituto Italiano di Tecnologia, Genova, Italy. [7]Bioinformatics and Computational Biology Program, Worcester Polytechnic Institute, Worcester, MA, USA. [8]Computer Science Department, Worcester Polytechnic Institute, Worcester, MA, USA. [9]Data Science Program, Worcester Polytechnic Institute, Worcester, MA, USA. [10]Physiology, Biophysics & Systems Biology Program, Weill Cornell Medicine, New York, NY, USA. [11]These authors contributed equally: Careen Foord, Andrey D. Prjibelski, Wen Hu. ✉e-mail: hut2006@med.cornell.edu

variables and their combinations. Using our technology of single-cell isoform sequencing (ScISOr-Seq)[26], we recently profiled single-cell isoforms in cerebellum at postnatal day 1 (P1)[26] as well as in pre-frontal cortex (PFC) and hippocampus at P7[27]. Additionally, multiple groups have tied alternative splicing to neurogenesis and differentiation[28,29] and splicing differences between neuronal subtypes have been linked to the distinct expression patters of the splicing factors *Nova*, *Rbfox*, *Mbnl*, and *Ptbp*[30]. Additionally, for the Brain Initiative, with an enhanced version of ScISOr-Seq (ScISOr-Seq2), we defined brain-region specific full-length isoforms at single-cell resolution, covering cerebellum, hippocampus, thalamus, striatum, and visual cortex at mouse postnatal day 56 (P56) and additionally mouse visual cortex and hippocampus at P14, P21, and P28[31]. Collectively, these data have revealed widespread cell-type specific isoform expression and brain-region specificity in isoforms for matched cell types—and dramatic developmental isoform changes in the mouse visual cortex for matched cell types. While certain markers in the cortex correlate with layer-specific location of cells, it is currently rarely possible to conclusively determine the layer in which an individual cortical neuron resides based solely on marker expression—especially in the human brain[32–34]. Therefore, in order to answer the question of which cortical layers are most strongly altered by splicing and poly(A)-site choice, technologies with spatial resolution are required. To this end, we and others have previously developed spatial isoform sequencing[27,35] which revealed spatially regulated isoform expression events. Many of these occurred at transitions from one brain-region to another, but others, like *Snap25*, occurred in a gradient throughout the mouse brain. At the employed spatial resolution of ~60 um, however, most physical spots covered multiple cells, thus hindering a cell-type specific view of spatially regulated isoforms. Additionally, our single-cell isoform investigation in human PFC[36] and hippocampus[31] have revealed many human alternative splicing events that cannot be modeled in mouse brain. Here, we address these limitations by developing spatial isoform sequencing (Spl-ISO-Seq) that improved the previous 60 μm to now 10 μm resolution. We coupled the spatial Curio Biosciences technology, a slide-based whole-transcriptome sequencing platform with high spatial resolution stemming from Slide-SeqV2[37], along with a 2-step protocol enriching for exon-containing and long cDNAs. We add a specialized software package for the analysis of spatially barcoded long reads (Spl-ISOquant). Using fresh frozen human visual cortex (Brodmann area 17) samples from pre-puberty donors aged 8–11 years ("children") to post-puberty (16–19 years, "young adults"; Y.A.), we investigate RNA biology regulation during this developmental period. Collectively, these approaches reveal widespread regulation of alternative exon inclusion, alternative acceptor usage, and poly(A)-site usage. Overall developmental RNA regulation in the cortex is stronger than in the white matter and among the cortical layers, layer 4 shows the highest regulation—an observation traceable to excitatory neurons. For oligodendrocytes, however, regulation is stronger in white matter than in cortical layers. Overall, we present a cell-type specific spatial transcriptomic technology optimized for long-read sequencing and identify region and cell-type specific splicing trends across human development.

## Results

### A spatial view of pre-puberty and post-puberty samples of the visual cortex

Our protocol for spatial isoform sequencing ("Spl-ISO-Seq") at near-single-cell resolution ensures that polyadenylated RNA molecules are spatially barcoded based on tissue placement on a barcoded slide. The workflow proceeds by producing spatially barcoded cDNA, which is split into two pools. (i) The first pool was fragmented for generating Illumina short reads for gene-expression estimation and cell-type deconvolution and (ii) the second pool for enrichment of exon-containing molecules, long-molecule selection, and long-read isoform

sequencing with Oxford Nanopore Technology (ONT) (Fig. 1a). We used this approach to compare a group of four human visual cortex samples from children (8–11 yrs) and four post-puberty young adults (Y.A.; 16–19 yrs; Supplementary Data 1). We sequenced ~245 million Illumina reads per individual. On average, these reads recovered 86.2% of expected barcodes (Supplementary Fig. S1). The visual system and specifically Brodmann area 17 (V1) are well studied in their architecture and function[38–40]. V1 is composed of 6 layers, which aid in visual signaling and processing. White matter is mainly composed of myelinated axons and oligodendrocytes. The layers of the visual cortex are distinguishable by key attributes, including cell density, layer thickness, and layer-specific markers. Illumina UMI counts per spot suggested the position of distinct cortical layers (Fig. 1b), which was confirmed with H&E stains of an adjacent tissue slice (Fig. 1c and Supplementary Fig. S2), short-read deconvolution methods[41] ("Methods"), and known layer-specific markers. These layer-specific markers were used to help identify layer cutoffs and were identified with Illumina sequencing as Illumina data was less sparse compared to ONT sequencing data (Supplementary Fig. S3). We also employed a cell-type deconvolution program[41], which identified which spots entirely contained individual cell types. Most cell-type deconvolution programs require large numbers of reads to accurately deconvolve cell types, thus Illumina sequencing was required for this process as ONT isoform sequencing lacked sufficient depth (Supplementary Fig. S4) and could not sufficiently identify layer patterns or cell types (Supplementary Fig. S5). Among spots classified as representing a single cell type ("singlets"), excitatory neurons were most abundant across all regions, followed by oligodendrocytes and vascular/endothelial cells (VENCs) (Fig. 1d, e and Supplementary Fig. S6). Out of all Illumina-identified spatial barcodes ($n = 482,777$), excitatory neurons had the highest rate of assignment (Supplementary Fig. S7a). Spots which were identified to contain two cell types ("doublets") were less common compared to assigned singlets and empty spots ("rejects"), and the majority consisted of excitatory neurons paired with another cell type (Supplementary Fig. S7b, c). Of note, 87.9% of all Illumina-identified spatial barcodes were also detected in ONT data. The barcodes which were highly sequenced by Illumina but not ONT were significantly upregulated in mitochondrial genes, which are not targeted by the exome enrichment (Supplementary Fig. S8). Comparing short-read gene expression between children and young adults, in agreement with the literature, we found increased expression of genes linked to long-term potentiation in the visual cortex in younger samples[42–44]. Conversely, and again agreeing with the literature[45,46], we found genes linked to oxidative phosphorylation to increase during puberty (Fig. 1f). In neurons, KEGG analysis revealed synaptic-vesicle-cycle genes and glutamatergic-synapse genes were upregulated. Likewise, glutamatergic-synapse genes as well as aldosterone-regulated sodium reabsorption were upregulated in astrocytes. As expected, oligodendrocytes also showed enrichment for myelin-related genes with a gene ontology analysis (Fig. 1g). Clustering analysis of short-read gene expression in excitatory neurons revealed that the strongest split represented the difference between white matter and cortex, with the second strongest split separating age groups rather than cortical layers. This observation supports the idea of robust cortical gene regulation during puberty (Fig. 1h). Performing principal component analysis, principal component 1 (PC1) and PC2 of singlets also demonstrate variability deriving from mitochondrial gene expression and cell-type differences predominantly between excitatory neurons and oligodendrocytes, which likely represent overall differences between cortex and white matter (Supplementary Fig. S9).

### A long cDNA enrichment approach enables improved spatial and single-cell isoform recovery

Long reads hold the promise to reveal the combination patterns of variable sites (for example, TSS, all splice sites, poly(A)-sites on

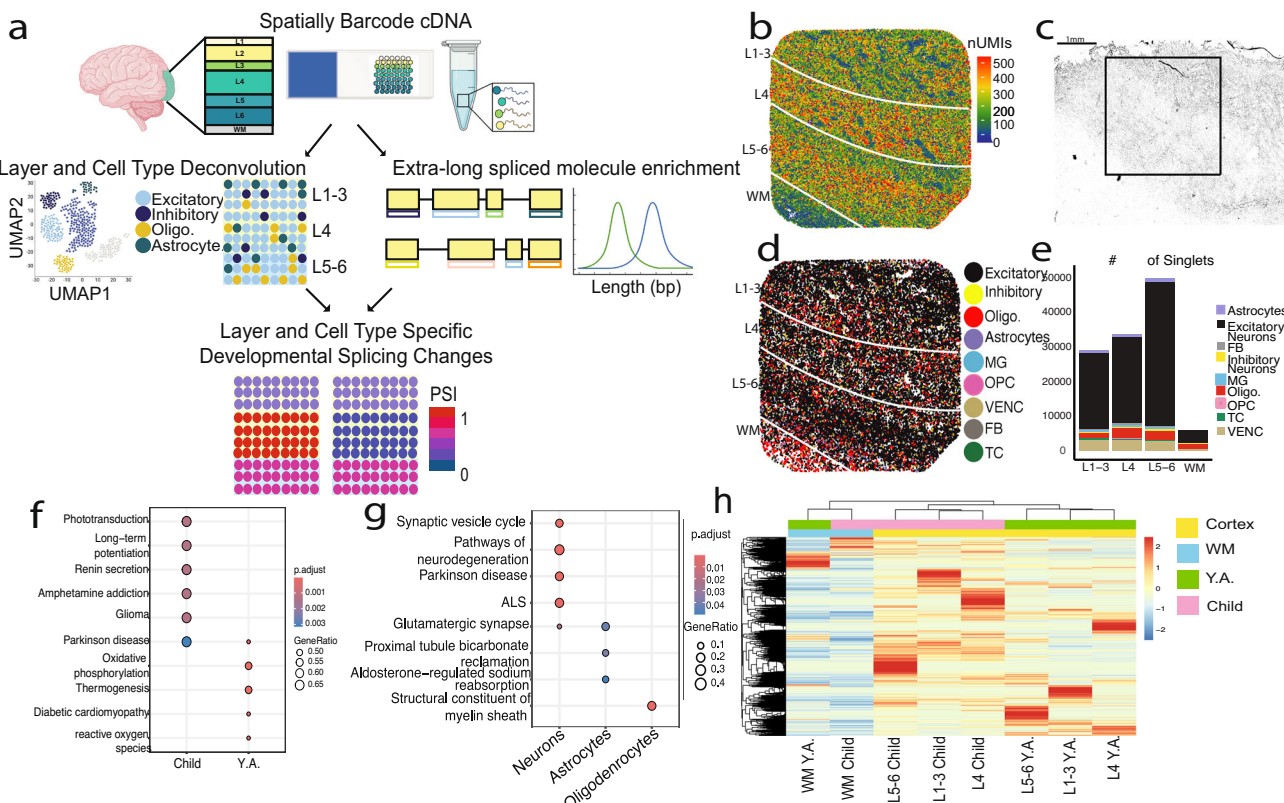

**Fig. 1 | Profiling of cell types and regions. a** Experimental overview. Spatially barcoded cDNA is separated into 2 pools: one which is short-read sequenced and used for layer and cell-type deconvolution, and the other which undergoes exome enrichment, long-molecule selection, and is long-read sequenced. The two sets of data are then combined to examine layer and cell-type specific developmental splicing changes. PSI indicates Percent Spliced In. Created in BioRender. Jarroux, J. (2025) https://BioRender.com/t5m4ory. **b** UMI count per spot plotted by spatial location on sample 1115. **c** Hematoxylin and Eosin stain on 10 µm-thick slice of tissue following experimental section of sample 1115 with approximate area captured in black square. Scale bar indicates 1 mm. **d** RCTD defined singlets plotted by cell type and spatial location. Cell types include excitatory neurons, inhibitory

neurons, oligodendrocytes (Oligo.), astrocytes, microglia (MG), oligodendrocyte precursor cells (OPC), vascular endothelial cells (VENC), fibroblasts (FB), and T cells (TC). **e** Number of singlets across layers (L1-6) and white matter (WM) for all samples combined. **f** KEGG enrichment of gene expression differences from short-read data when grouping by age groups, child (8–11) and young adult (16–19). **g** KEGG enrichment of neuron and astrocyte singlets and GO enrichment for oligodendrocytes. Enrichment analyses were performed with the clusterProfiler R package ("Methods") with a one-tailed hypergeometric test followed by a BH correction. **h** Heatmap of short-read gene expression in excitatory neurons by brain area (Layer 1–6 and WM) and age group. Y.A. indicates Young Adult.

RNAs)[21]. However, cDNAs produced by spatial approaches or single-cell DNAs are commonly short, often due to truncation of full transcripts. In the case of Curio Bioscience's library preparation, a random hexamer rather than a 5′ TSO molecule is employed to generate 2nd strands. Thus, most molecules are not complete as the library preparation is not designed to capture full-length molecules, but rather only gene expression. We hence engineered cDNA amplification and purification methods to enrich for longer, spliced cDNAs. After isolating spatially barcoded cDNA following the Curio Bioscience's pipeline, exon-spanning cDNA was enriched for using Agilent exome enrichment probes ("Methods"), which removes purely intronic molecules and enriches for exonic ones. Additionally, the PCR following exome enrichment was stopped at cycle 8 and cleaned up with 0.48× SPRIselect beads in 1.25 M NaCl-20% PEG buffer to yield long cDNA. We then performed another PCR such that only a longer set of the cDNAs was amplified, followed by another cleanup ("Methods"). We optimized Spl-ISO-Seq using two PFC samples (one child and one young adult) (Supplementary Fig. S10). Using the Curio spatial-genomics approach and naïve long-read sequencing, we see a median read length of 502 bp ("Standard"). Employing exome-enrichment probes to remove intronic cDNAs left this number largely unchanged ("Standard Exome"). Our new protocol, involving exome enrichment and long-cDNA enrichment ("Long Exome"), increases the median read length 2.7-fold to 1358 bp (Fig. 2a; Standard: Mean = 775.61, Standard

Deviation (SD) = 941.65; Standard Exome: Mean = 725.53, SD = 455.23; Long Exome: Mean = 140.45, SD = 673.90). The Long Exome dataset contained a slightly higher percentage of intron-retained reads compared to the other two protocols (Supplementary Fig. S11). For spliced reads, however, the exome enrichment step yields a dramatic increase in spliced reads (Fig. 2b). Taken together the long protocol increases the median exon number per spliced read from 4 to 5 (Fig. 2c; Standard: Mean = 4.37, SD = 1.78; Standard Exome: Mean = 4.41, SD = 1.67; Long Exome: Mean = 5.84, SD = 2.90). Furthermore, comparing the number of reads assigned to each gene between Standard and Long Exome experiments yielded a high correlation (Fig. 2d; Pearson's $R = 0.834$, $p < 2.2e\text{-}16$). Similarly, transcript expression is also correlated between the two datasets (Supplementary Fig. S12a, Pearson's $R = 0.69$, $p < 2.2e\text{-}16$) and transcript expression is not related to transcript length (Supplementary Fig. S12b). Thus, the enrichment for long fragments does not strongly shift gene expression patterns when comparing two exome enriched samples. Additionally, performing the exome enrichment itself does not significantly alter gene expression compared to the standard preparation (Supplementary Fig. S13; Pearson's $R = 0.814$, $p < 2.2e\text{-}16$). Not observing a strong shift in gene expression profile yet having longer reads could be explained by covering a larger fraction of the isoforms to which the molecules are assigned. We therefore calculated the fraction of bases of an annotated transcript covered by a read after the read was assigned to this

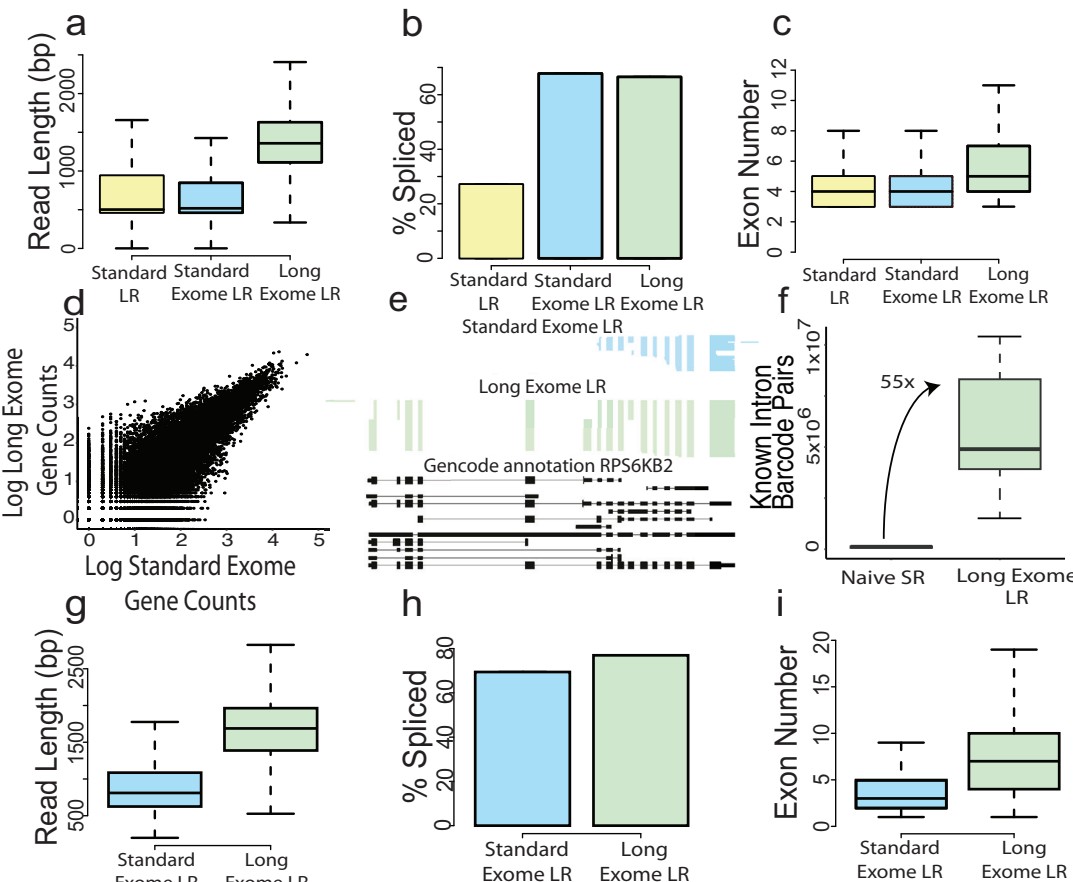

**Fig. 2 | A long cDNA enrichment approach enables improved spatial and single-cell isoform recovery. a** Read length of ~100,000 long reads randomly sampled from each spatial dataset ($n = 2$). Standard LR: cDNA from standard protocol prior to tagmentation; Standard Exome: cDNA from standard protocol, which is enriched for exonic reads; Long Exome: cDNA from standard protocol, which is enriched for exonic reads and longest molecules. **b** Percent of spliced molecules from each group. **c** Number of exons per read in barcoded, mapped, and spliced reads from 2 samples per group. Standard LR: $n = 1719458$; Standard Exome LR: $n = 4396753$; Long Exome LR: $n = 1686605$ reads. **d** Correlation of counts per gene of a Standard Exome dataset compared to the Long Exome dataset from the same spatial slide ($n = 1$ sample per group). Counts are log10 transformed. **e** ScisorWiz[48] plot of the gene RPS6KB2. Each horizontal line is a sequenced read from each respective dataset, which come from the same slide. Gencode annotated transcripts are plotted in black. **f** Total number of intron-barcode pairs from naive Illumina sequencing compared to Long Exome sequencing from all samples ($n = 8$). **g** Read length of 10,000 reads randomly sampled from each single-cell dataset ($n = 1$). **h** Percent of spliced molecules per group. **i** Number of exons per read in barcoded, mapped, and spliced reads (Short Exome $n = 8110525$, Long Exome $n = 407764$). All Boxplots show the median (center line), upper bound at 75th percentile, lower bound at 25th percentile with whiskers at $1.5 \times$ interquartile range (IQR). Color indicates method, yellow is from Standard LR, blue from Standard Exome LR, and green from Long Exome LR.

transcript. Using the Long Exome approach, this fraction increased from 0.49 to 0.70 and from 0.53 to 0.59 in the two samples, respectively (Supplementary Figs. S14 and S15a, b). Of note, even in cDNAs of bulk full-length molecules[47] the resulting fraction of transcript coverage is 0.89. The difference between sequencing methods becomes even more pronounced when only examining spliced reads (Supplementary Fig. S15c, d). Overall, the Long Exome approach allows for the capture of larger portions of transcripts, which is greatly increased in longer transcripts compared to the Standard LR and Standard Exome LR (Supplementary Fig. S15e–h). Importantly, a truncation simulation of ONT reads suggests that isoform assignment recall and precision are affected by the coverage fraction per transcript (Supplementary Data 2). Thus, enrichment for longer molecules both tends to cover more of the original RNA isoforms that the reads represent, as well as maximizes the accuracy of isoform assignment and subsequent quantification. An example of this higher fraction of transcript bases covered can be found in the *RPS6KB2* gene. Before selection of long cDNAs, the sequenced reads cover an average of 5.4 exons per read, while with selection of long cDNAs, the sequenced reads cover an average of 12.3 exons per read and usually reach the first annotated

exon[48] (Fig. 2e). We then explored whether short-read analysis alone could serve for splicing analysis in our brain slices. To this end, we counted the number of detected splicing events per barcode and added these from all barcodes, which we termed intron-barcode pairs. For the short-read experiment, we found an average of 108,343.6 events per sample, while we found 5,910,532 for the Spl-ISO-Seq protocol—a 55-fold increase (Fig. 2f; Naive Short Read (SR): SD = 39672.25; Long Exome Long Read (LR): SD = 3242296). Of note, spliced short reads showed dramatically lower gene coverage compared to both naïve short read and Long Exome LR sequencing (Supplementary Fig. S16). Although Naïve Short read data showed high depth, gene coverage was limited to the regions near the Poly(A) site. In contrast, Long Exome LR data showed moderate, although more uniform coverage. We then tested whether the selection for long molecules had equally positive effects for our previously published single-cell experiments. Indeed, the long protocol has even more drastic increases in a separate, mouse single-cell sequencing experiment: the read length is increased from a median of 810 to 1688 bp (both after exome enrichment, Fig. 2g; Standard Exome: Mean = 919.83, SD = 463.88; Long Exome: Mean = 1701.81, SD = 592.45). While the spliced-read

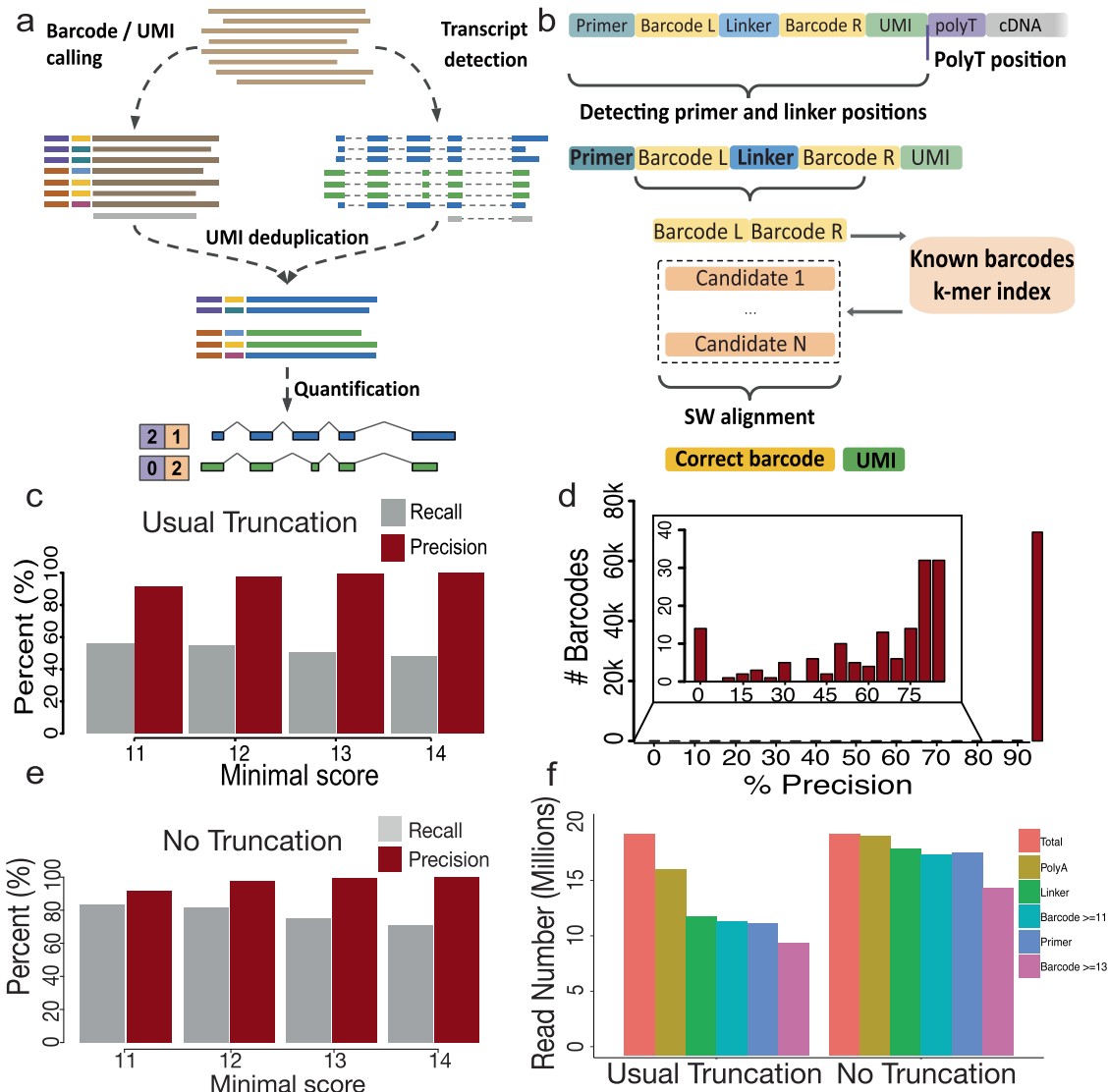

**Fig. 3 | Spl-ISO-quant enables long-read analysis including highly specific barcode deconvolution. a** Outline of Spl-IsoQuant algorithm. **b** Barcode calling and UMI determination in Spl-IsoQuant. **c** Precision and recall as a function of minimal scores overall in Spl-IsoQuant with truncation, which is representative of ONT data. Grey indicates recall and red indicates precision. **d** Precision for individual barcodes. **e** Precision and recall as a function of minimal scores overall in Spl-IsoQuant with no truncation. Grey indicates recall and red indicates precision. **f** Number of reads found by Spl-IsoQuant to meet criteria cutoffs separated by reads modeled with usual truncation and no truncation.

fraction increases only slightly (Fig. 2h), the median exon number per read rises from 4 to 8 (Fig. 2i; Standard Exome: Mean = 4.98, SD = 2.46; Long Exome: Mean = 8.19, SD = 3.94). This demonstrates that this new protocol functions well on cDNA derived from multiple protocols and regions. Taken together, these findings show that both steps of our new protocol, namely exome enrichment and long-cDNA isolation are required for yielding spliced reads derived from transcripts of different sizes.

## Spl-ISO-quant enables long-read analysis including highly specific barcode deconvolution for spatial experiments with near-single-cell resolution

We have recently published IsoQuant and shown that it is highly specific in terms of isoform detection and read-to-isoform assignment for both Oxford Nanopore and PacBio long-read experiments[49]. IsoQuant has recently been shown—without our involvement—to exhibit very strong performance. Specifically, the authors found IsoQuant as "a highly effective tool for isoform detection with LRS, with Bambu and StringTie2 also exhibiting strong performance"[50]. Here, we added barcode deconvolution and UMI deduplication, as well as isoform quantification (Fig. 3a). While these problems were previously addressed by various tools[51–54] (BLAZE, SiCeLoRe, FLAMES, and scna-noseq), all of these methods are designed specifically for 10× Genomics long-read data and cannot be easily adapted for other protocols with a distinct molecule structure, such as Curio. Furthermore, Curio has about tenfold more barcoded spots in a single experiment compared to a typical 10× Genomics single-cell experiment. Additionally, Curio features barcodes of 14 bp versus 16 bp long barcodes for 10×. Thus, the Curio data is characterized by substantially higher barcode collision probability (roughly 100-fold), which even further complicates the barcode demultiplexing process. Using the known barcode structure of the Curio protocol, we first detect PCR primer and barcode linker positions in the vicinity of poly(A)-tails. The known barcodes that come with each of the purchased slides are exploited to build a k-mer index, which allows to quickly obtain a limited number of candidate barcodes for every read. For these candidates a

Smith–Waterman alignment is employed to assign a score to each barcode candidate ("Methods"). The maximal score of 14 represents 100%-identity of the long-read derived barcode candidate and a barcode in the prior list. This score is then used to determine the correct barcode, and the position of the barcode yields the candidate sequence of the UMI (Fig. 3b; "Methods"). Subsequently, barcoded, spliced, and UMI filtered reads are used for downstream splicing analysis, where UMI filtering does not introduce any significant bias with respect to read length or splice sites (Supplementary Fig. S17 and Supplementary Data 3). We performed simulation experiments, introducing errors into in-silico reads according to an error profile learned from Oxford Nanopore alignments, including read truncation, which is a commonly occurring phenomenon in ONT sequencing ("Methods"). We used these experiments to determine precision and recall of our barcode-calling strategy. Allowing barcode calls with a minimal score of 11, yielded a precision of 91.6%. This number rose to 97.3% (score = 12), 99.5% (score = 13), and 99.9% (score = 14). As expected, recall also decreased with higher score requirements from 55.2% (score = 11) down to 50.4% for score 13 (Fig. 3c). Given the small precision difference between score = 13 and score = 14, we used score = 13 in the following analyses. We also verified if specific barcodes with score = 13 could be more problematic than others. Overall, close to all barcodes showed a precision of ≥95%. However, a very few barcodes (118 of ~70 k) showed precision <85% and 14 barcodes precision of 0–5%. These few barcodes contribute a small fraction of reads and thus do not affect slide-wide conclusions. However, if any statements are made for a very small number of barcodes (≤10), our results indicate that such barcodes need to be tested for their individual precision to avoid false positive results derived from few barcodes (Fig. 3d). Of note, the above simulations included substitutions, insertions and deletions as well as read truncation. The former three error sources in principle still allow for reads covering the entirety of the barcode—a situation, in which errors can potentially be remedied by state-of-the-art algorithms. Read truncation however, can completely delete a barcode—a situation that no algorithm could ever remedy. To understand the influence of read truncation, we performed simulation experiments without read truncation. This left precision largely unchanged, but increased recall to 83.3% (score = 11), 81.8% (score = 12), 75.4% (score = 13), and 70.9% (score = 14). Thus, a large portion of the recall loss is due to read truncation (Fig. 3e). In the simulation scenario with truncation, the largest numbers of reads were lost when asking for the presence of the poly(A) tail and/or the linker in between the two barcodes. This loss did not occur in the simulation without truncation (Fig. 3f). In summary, Spl-IsoQuant enables highly precise barcode recognition and pinpointing of problematic barcodes. The strongest contributor to not finding barcodes is read truncation—a chemical occurrence that cannot be remedied by algorithms.

## Developmental changes in splicing affect cortical layers more than white matter

The architecture and function of the visual system and V1 are well studied in the neuroscience field[38–40]. The visual cortical layers are distinguishable by key attributes including cell density, layer thickness, and layer-specific markers. We used H&E stains (Supplementary Fig. S2a–h) and short-read deconvolution (Supplementary Fig. S18a–h) to identify cell densities and layer thickness and examined the location of layer-specific marker expression. Based on these measurements, we divided each slide into layers 1–3 ("L1–3"), layer 4 ("L4"), layers 5–6 ("L5–6"), and white matter ("WM"). In addition to the previously mentioned two PFC samples, we long-read sequenced eight visual cortex samples to yield four male children individuals (8–11 yrs) and four male young adults (16–19 yrs) with ONT (Supplementary Fig. S19a–f and Supplementary Data 4). Intron lengths were in-line with our previous results, in that close to no expressed introns were found below 67 bases. Internal-exon length (average = 143 bases) and

terminal-exon length (average = 642 bases) were both in-line with recent literature[55] (Supplementary Fig. S20a, b). As we established previously[56,57], few introns are <70 bases and few introns extended past 20 kb (Supplementary Fig. S20c, d). Overall, barcoded molecules assigned to a transcript identified 19,686 annotated genes with ≥10 UMIs. Downsampling experiments showed some signs of saturation, although perfect saturation was not attained (Supplementary Fig. S20e). Similar trends are found when repeating experiments with ≥3 UMIs (Supplementary Fig. S20f).

We first tested individual exons for distinct exon inclusion between children and young adults ("Methods"). An alternative exon is included or excluded in a cDNA molecule and here we consider two age groups—leading to a 2 × 2 contingency table[3,27,36]. Of note, we found many exons with a change in exon inclusion values (otherwise known as deltaPSI or $\Delta\Psi$) close to −1 or +1 in the cortical layers, implying complete developmental shifts in exon inclusion (Fig. 4a). In white matter, we observed much fewer of such extreme shifts (Fig. 4b). In the cortical layers, 18.54% of tested exons changed inclusion during puberty, while only 6.14% did so in white matter (Fig. 4c and Supplementary Fig. S21). Of note, performing the same analyses with spliced short reads resulted in a 93.3-fold decrease in cortical tested exons, where not a single exon was identified to be regulated across age after FDR correction (Supplementary Fig. S22). Considering $\Delta\Psi$ values of significant exons in both compartments, we found those in the white matter to exhibit negative (that is decreasing inclusion with developmental time) changes more often than those in the cortical layers—the latter of which showed positive and negative changes at a more balanced ratio (Fig. 4d). Importantly, larger spot numbers and read numbers per spot can lead to higher significance numbers in one of the areas. To test whether this affected cortex and white matter differentially, we performed downsampling experiments ("Methods") that equalized data imbalances. These downsampling experiments confirmed that indeed cortical layers had a higher rate of developmentally-regulated exon inclusion than the white matter did (corrected 2-sided Wilcoxson rank sum test; FDR = $1.1 \times 10^{-11}$, Fig. 4e; Cortex: Mean = 0.55, SD = 0.80; WM: Mean = 0.0, SD = 0.0). We additionally performed simulations to examine how read number and $|\Delta\Psi|$ contributed to false positive and negative exons detected (Supplementary Fig. S23a). This analysis determined that 10–19 reads per comparison is necessary to detect a change in exon inclusion, which aligns with the results (Supplementary Fig. S23b). An example of alternative-splicing regulation specific to the cortical layers is the *SLC12A2* gene, in which an exon increases inclusion from 19.3% to 70.2% in cortex across age. In white matter however, we found no significant change (Fig. 4f). This exon keeps the reading frame and has been shown to impact sodium influx by the cell[58]. Searching for SynGO[59] enrichment of genes with significant developmental exon inclusion changes in cortical layers yielded strong enrichment for "postsynaptic density", "synaptic vesicle cycle" and "receptor localization" in location or function mode of SynGO (Fig. 4g, h). However, similar analysis for genes with altered exons in the white matter gave no such enrichment (Supplementary Fig. S24). Thus, developmentally regulated splicing in cortical layers is heavily connected to the postsynaptic density. Next, we identified exons which contain start codons and coding sequences based on published annotations. Cortical developmentally regulated exons (FDR < 0.05 & $|\Delta\Psi|$ > 0.2) contained start codons less often than exons that were not developmentally regulated (FDR > 0.05 & $|\Delta\Psi|$ < 0.05; corrected 2-sided Fisher test, FDR = $1.19 \times 10^{-8}$). Thus, there is a tendency not to alter translation initiation in postnatal development of the cortical layers. In the white matter however, we did not observe such a difference between developmentally-regulated and non-regulated exons (corrected 2-sided Fisher test, FDR = 0.39; Fig. 4i). On the other hand, developmentally regulated exons in the cortical layers contained non-coding sequence significantly more often than non-regulated exons—an observation that we could again not make in

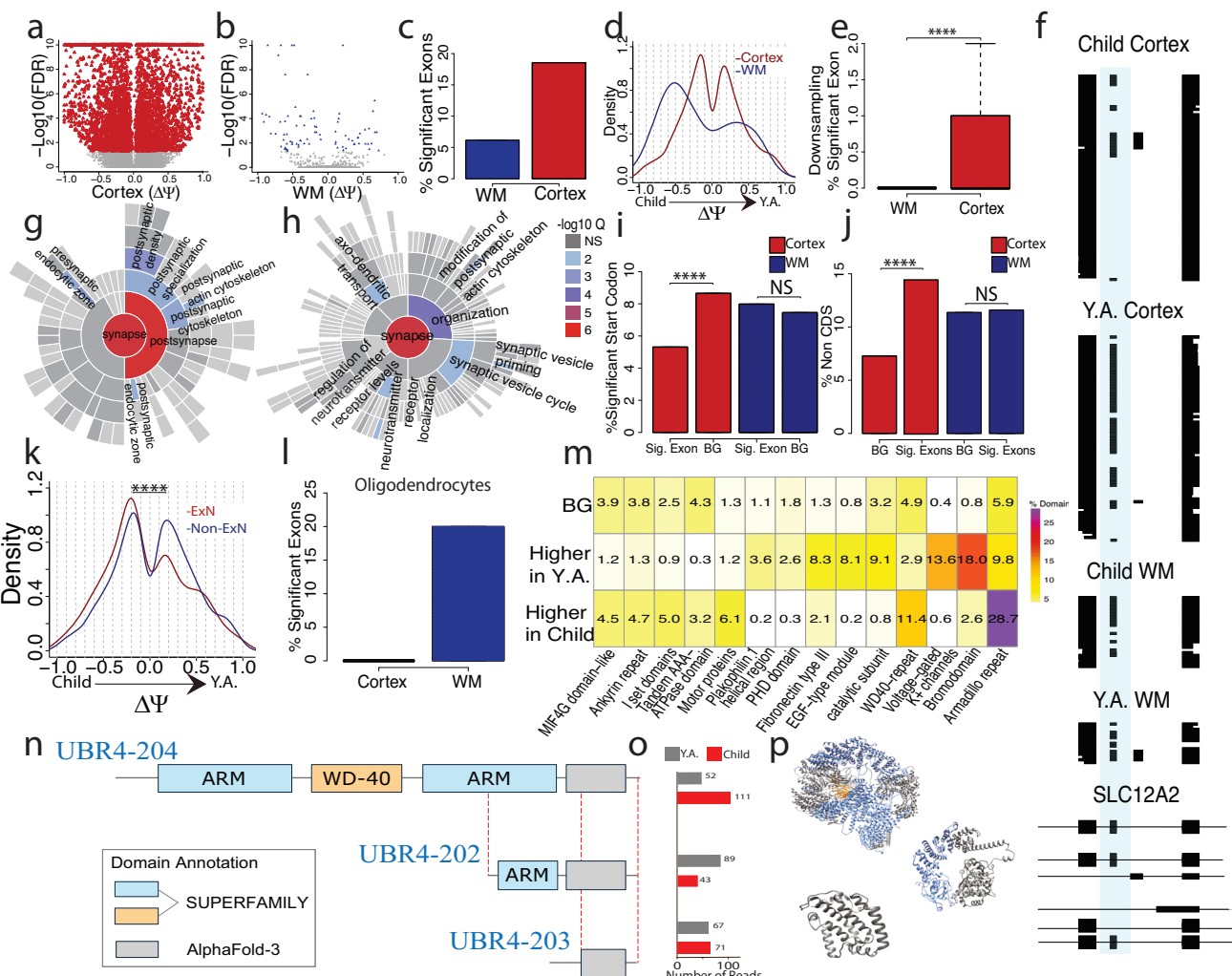

**Fig. 4 | Developmental changes in splicing affect cortical layers more than white matter. a** Exons tested between ages in cortex by deltaPSI (YA-Child) and -log10(FDR). Red points are significant exons and grey points are non-significant. **b** Exons tested between ages in WM by deltaPSI and -log10(FDR). Blue points are significant exons and grey points are non-significant. **c** Percent significant of tested exons and |deltaPSI| ≥ 0.2 for cortex and WM. Cortex percent significant: 18.54%; WM percent significant: 6.14%. **d** deltaPSI density of significant exons for cortex and WM. **e** Percent significant exons from downsampling experiments where age groups (n = 4 Child, n = 4 Y.A.) are combined and compared. Exon selection and percent significant calculations are repeated 100× per region and distribution of percent significant is plotted ("Methods", n = 100). Boxplot shows the median (center line), upper bound at 75th percentile, lower bound at 25th percentile with whiskers at 1.5 × IQR. **f** ScisorWiz[48] plot of exon chr5_128177105_128177152_+ in *SLC12A2* gene. **g** SynGO location enrichment of significant cortex genes. **h** SynGO function enrichment of significant cortex genes. **i** Percent of exons associated with start codons between highly significant and a background set of exons. **j** Percent of non-CDS exons between highly significant and a background set of exons. **k** deltaPSI density of significant exons of excitatory neurons and non-excitatory neurons. **l** Percent significant of tested exons and |deltaPSI| ≥ 0.2 for cortex and WM in Oligodendrocytes. Cortex percent significant = 0%; WM percent significant = 20%. **m** Enrichment of protein domains in cortex highly significant alternative exons (|deltaPSI| > 0.5 and FDR < 0.05). **n** Diagram of 3 transcripts of the multidomain protein UBR4, which affect its domain architecture. Transcripts contain 2 Armadillo repeats (ARM) and 1 WD-40 domain (UBR4-204), 1 shorter ARM repeat only (UBR4-202), or neither domain types (UBR4-203). **o** Number of spliced and barcoded reads assigned to each transcript, separated by Y.A. and Child. **p** AlphaFold3 predictions of protein structure; blue = ARM repeats and yellow = WD-40.

the white matter (corrected 2-sided Fisher test, cortex FDR = 2.40 × 10⁻²²; Fig. 4j). We next considered exons which are developmentally regulated in the cortex. Exploiting the near-single-cell nature of this spatial technology and short-read gene expression, we classified each physical spot as either excitatory neuron, inhibitory neuron, oligodendrocyte, astrocyte, microglia, vascular cell, fibroblast, or unclassifiable (Supplementary Fig. S18). Unclassifiable spots usually have low RNA content, either because of biological low expression, such as a cell-free area or because of technical limitations. Separating the most abundant cell type, excitatory neurons, from other spots, we found developmentally regulated exons in excitatory neurons to show a strong trend for decreasing their inclusion with increases in age, while exons from other spots showed more balanced proportions of increasing and decreasing proportions (2-sided Wilcoxon rank sum

test, $p = 9.44 \times 10^{-9}$; Excitatory Neuron (ExN): Mean = -0.04, SD = 0.43; Non-ExN: Mean = 0.03, SD = 0.41; Fig. 4k). Considering the second most abundant cell type—oligodendrocytes—we asked whether WM oligodendrocytes had splicing changes more frequently than cortical oligodendrocytes. In opposition to the trends seen in all cortical and white matter spots, we found WM oligodendrocytes to show splicing changes more frequently than cortical oligodendrocytes (Fig. 4l). Sufficient data was not available to down-sample this result. However, WM overall has a smaller number of assigned oligodendrocytes than in the combined cortical layers, suggesting that this result is not due to power imbalances. We then searched for protein domains that are affected by cortical developmentally regulated exons and considered the percentage of exons affected by age. Exons were grouped based on which age group they were most included (higher in children or higher

in Y.A.). As a background set, we considered exons without any apparent regulation ("Methods"). After a Fisher test of higher-young-adult inclusion and higher-children inclusion exons and a Bonferroni correction, 14 protein domains showed differences. Some of these groups affected few exons overall, but others had higher differences. For example, voltage-gated potassium channels were present in 13.6% of higher-adult-inclusion exons but only in 0.64% of higher-children-inclusion exons. Thus, voltage-gated potassium channels are increasingly used in early adulthood (as compared to younger children) and a similar observation was made for bromodomains. Conversely, Armadillo repeats (ARM) were frequently observed in higher-children-inclusion exons (Fig. 4m). ARM domains is an example of a ubiquitous structural fold of alpha solenoid consisting of a varying number of stacked short helices[60]. *UBR4* is a gene that contains two ARM domains and has been identified as one of the key proteins in the mammalian brain associated with neurogenesis, neuronal migration, and neuronal signaling[61]. However, the role of AS in regulating its functional activity has not yet been studied. We found that *UBR4-204* has both ARM domains, where others either have both domains missing (*UBR4-203*), or a shortened copy of one ARM domain (*UBR4-202*), which could indicate the absence or substantial reduction of its function (Fig. 4n). Interestingly, *UBR4-204* reads are increased in the child cortex compared to young adults. That trend is flipped in the *UBR2-202* where reads are increased in young adults rather than children (Fig. 4o). Due to the modular nature of the ARM domains, consisting of short helices of the same length, it is expected that the shorter version of the ARM domain will fold into a stable structure even though shorter proteins are generally more unstable ("Methods", Fig. 4p). The observed stability can be explained by the previously identified property of ARM repeats to self-assemble even when they are non-covalently bound[62]. Furthermore, the modular stability of ARM, TRP, and other repeat proteins has increasingly been utilized in the modular design of synthetic repeat proteins[63].

## Developmental polyadenylation regulation equally affects cortical layers more than the white matter

Our assay equally reveals changes in poly(A) sites. We tested whether poly(A) sites would follow similar trends as alternative exons using our concept of poly(A)-site tests[27,31]. In contrast to alternative exons with two possible states, a gene can have many poly(A) sites. We therefore employ $2 \times N$ tests for poly(A)-site testing[27]. Similarly, as alternative exons, we found a higher fraction of developmentally-regulated genes (estimate = 24.67) in cortical layers as compared to white matter spots (estimate = 3.70). However, while we had observed a threefold higher value for alternative exons between cortical layers and white matter, for poly(A)-site choice we observed a $a >$ sixfold higher level (Fig. 5a). In contrast to $\Delta\Psi$ values for alternative exons, the deltaPI($\Delta\Pi$) values for poly(A) sites are, by definition, positive. For genes with significant poly(A)-site regulation, these values were stronger for the white matter, which is likely a consequence of lower read numbers requiring stronger effect sizes to achieve significance (Fig. 5b; Cortex: Mean = 0.25, SD = 0.20; WM: Mean = 0.29, SD = 0.17). Downsampling experiments again showed that the observation of more poly(A)-site regulation in the cortex than in the white matter remains true when equalized statistical power is enforced (2-sided Wilcoxon rank sum test; $p = 8.12 \times 10^{-38}$; Cortex: Mean = 7.92, SD = 4.48; WM: Mean = 0, SD = 0; Fig. 5c). An example of this is the *VBP1* gene, in which the usage of a downstream poly(A) site is dramatically increased with age in the cortex, while in white matter the downstream site is exclusively used regardless of age (Fig. 5d). Poly(A)-site choice defines the length of the transcript's last exon. We separated genes into 2 groups based on those which lack developmentally regulated Poly(A)-sites ($|\Delta\Pi| < 0.2$, "non-changing") and those which undergo developmentally regulated Poly(A)-sites ($|\Delta\Pi| > 0.2$, "changing"). We defined these groups based on $|\Delta\Pi|$ criteria rather than significance in order to limit the number of

false positives while also capturing substantial changes which may lack sufficient power to reach significance. We first measured last exon length in non-changing tested genes ($|\Delta\Pi| < 0.2$; "Methods") both in the child and young adult cortex—and similarly in the white matter. No significant differences in length were detected between regions and age groups (Fig. 5e; WM Child: Mean = 443.34, SD = 303.42; WM Y.A.: Mean = 404.67, SD = 278.49; Cortex Child: Mean = 431.52, SD = 291.08; Cortex Y.A.: Mean = 444.89, SD = 316.33). However, genes with poly(A)-site regulation ($|\Delta\Pi| > 0.2$) in development had shorter last exons in the white matter than in cortex—regardless of age group (corrected 2-sided Wilcoxon rank sum tests; cortex Y.A. v WM Y.A. FDR = $4.96 \times 10^{-5}$; cortex Y.A. v WM child FDR = $1.36 \times 10^{-3}$; WM Y.A. v cortex child FDR = $1.31 \times 10^{-5}$; WM child v cortex FDR = $4.52 \times 10^{-4}$; WM Child: Mean = 326.57, SD = 198.45; WM Y.A.: Mean = 315.09, SD = 197.41; Cortex Child: Mean = 426.70; SD = 278.65; Cortex Y.A.: Mean = 428.03, SD = 294.29). Thus, white matter poly(A)-site regulation affects genes with short last exons (Fig. 5f). An example of this length observation can be seen in the *KAT8* gene (Supplementary Fig. S25). Additionally, a similarly to the length of last exons, the UTR length of genes with regulated poly(A) sites is shorter in WM compared to cortex (Fig. 5g). We also found that the genes which exhibit developmentally regulated exon inclusion ($|\Delta\Psi| > 0.2$) also tend to show developmentally regulated Poly(A) site usage in the cortex (2-sided Fisher test $p = 1.49 \times 10^{-16}$; Fig. 5h), indicating potential coordination between these processes.

Of note, although many of the total barcoded, spliced reads have an annotated poly(A) site, only 6.35% have a TSS, making the ability to do full-isoform tests challenging. By employing our long-molecule capture, we have increased the length and % transcript covered, but still lack a majority of full-length cDNAs. Even so, we performed full-length isoform tests on the full-length reads and confirmed that the number of significant genes with alternative-isoform usage per comparison is lower than on an individual exon basis (Supplementary Data 5). When comparing the child and young adult cortex, we identify 266 genes which exhibit alternative isoform usage between age groups. Of these 266 genes, 90 contain significantly altered exons from our primary analysis, where 30 genes are associated with two or more. Additionally, we find similar trends to alternative exons and Poly(A) sites demonstrating that cortex contains more significant isoform alternative usage than WM across age (Cortex: 25.32%, 95%-confidence intervals [13.40, 37.24]; WM: 2.78%, 95%-confidence intervals [1.70, 3.85]; Supplementary Fig. S26a), although we note that these values are not downsampled. An example of alternative-isoform usage in cortex can be found in M-Phase Phosphoprotein 6 (*MPHOSPH6*), a gene which is involved in the RNA exosome process[64]. In this example, one isoform is primarily used in the child cortex, whereas another isoform is primarily used in the young adult cortex (Supplementary Fig. S26b). Thus, Spl-ISO-Seq enables the capture of full-length isoforms and the identification of differences in isoform usage, albeit at a sparser rate than at the individual-exon level.

## Among cortical layers, layer 4 shows the strongest splicing alterations

We divided the cortex into three well-distinguishable compartments: Layers 1–3 ("L1–3"), layer 4 ("L4"), and layers 5–6 ("L5–6"). We found many altered exon-inclusion events in each compartment between age groups (Fig. 6a–c). Testing for differentially spliced exons, L4 had the highest fraction of developmentally regulated exons: 13.57% of tests were significant for L1–3, 16.12% for L4, and 12.45% for L5–6 (Fig. 6d). L1–3 and L4 showed approximately similar trends of up and down-regulated exons. Surprisingly, however, L5–6 revealed a markedly different trend: alternative exons were frequently downregulated after puberty (Fig. 6e; L1–3: Mean = 0.03; SD = 0.44; L4: Mean = 0.03; SD = 0.42; L5–6: Mean = −0.04, SD = 0.43). Downsampling experiments also enforcing that no single or pair of individuals can dominate the data for a given exon, showed that this observation is robust to equalizing

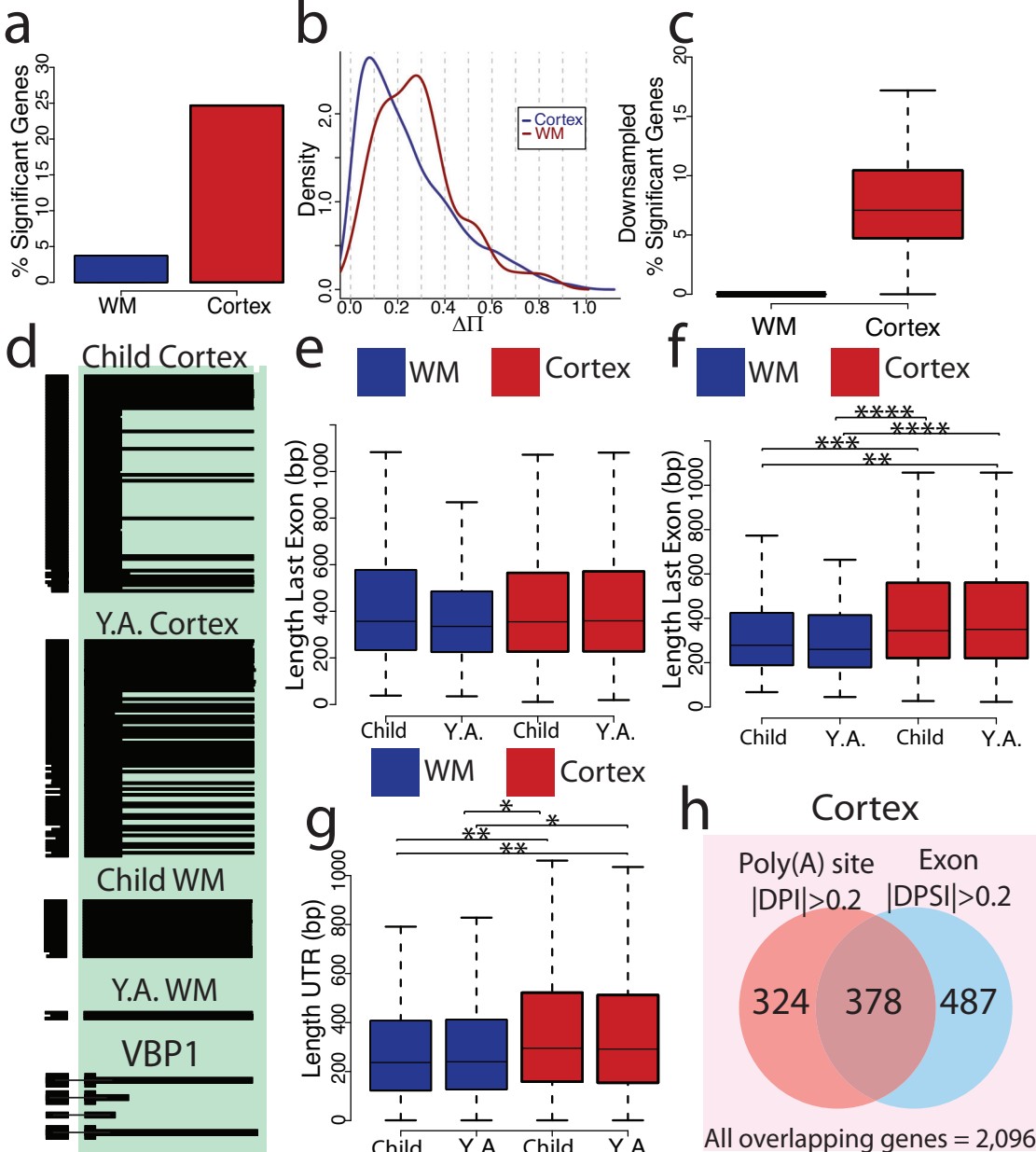

**Fig. 5 | Developmental polyadenylation regulation equally affects cortical layers more than the white matter. a** Percent significant genes with differential polyA sites. Cortex percent significant: 24.67%, WM percent significant: 3.70%. **b** PolyA deltaPI density in cortex and WM. **c** Downsampling experiments of percent significant genes with alternative PolyA sites where age groups ($n = 4$ Child, $n = 4$ Y.A.) are combined and compared. Sampling considered individuals equally and selected 20 reads and 50 genes randomly. Genes were resampled 100× and calculated percent significant per iteration ("Methods", $n = 100$). **d** ScisorWiz[48] example of alternative poly(A) site in the VBP1 gene. **e** Average length of last exons per tested genes with |deltaPI| < 0.2 compared across regions and age groups. Age groups combine 4 samples each. Child WM: $n = 595$; Y.A. WM: $n = 643$; Child Cortex: $n = 2350$; Y.A. Cortex: $n = 2250$. **f** Average length of last exons per genes with |

deltaPI| > 0.2 compared across regions age groups. Age groups combine 4 samples each. Child WM: $n = 152$; Y.A. WM: $n = 161$; Child Cortex: $n = 478$; Y.A. Cortex: $n = 470$. **g** Average length of UTR per genes with |deltaPI| > 0.2 compared across regions age groups. Age groups combine 4 samples each. Child WM: $n = 386$; Y.A. WM: $n = 392$; Child Cortex: $n = 552$; Y.A. Cortex: $n = 551$. **h** Overview of all cortical overlapping genes that were tested in both Poly(A) and exon tests ($n = 2096$). Genes were classified as only having |deltaPI| > 0.2, only having a |deltaPSI| > 0.2, or both. **** = $P \le 0.0001$, *** = $P \le 0.001$, ** = $P \le 0.01$, * = $P \le 0.01$. Boxplot center line is at the median, upper bound at the 75th percentile, lower bound at the 25th percentile with whiskers at 1.5 × IQR. WM is colored in blue and Cortex is colored in red. A two-sided Wilcoxon rank-sum test was applied to all the comparisons shown in (**c, e–g**).

statistical power (corrected 2-sided Wilcoxon rank sum test; L1–3 V L4 FDR = 0.0024; L4 V L5–6 FDR = 0.0037; L1–3: Mean = 0.33, SD = 0.53, L4: Mean = 0.7, SD = 0.82; L5–6: Mean = 0.39, SD = 0.68; Fig. 6f). As an example, three children show strong exon inclusion, while three young adults exhibit mostly exon exclusion in layer 4 of the cortex in the *TSPAN19* gene (Fig. 6g, h). SynGO analysis revealed no significant enrichment of any synaptic compartments in altered exon inclusion in L1–3 (Supplementary Fig. S27a). However, in L4 developmentally regulated splicing, we found enrichment of the postsynaptic compartment and to a lesser degree as well in L5–6 (Fig. 6i and Supplementary Fig. S27b). Similarly to observations on alternative exons (see Fig. 5 and prior panels of Fig. 6), we found L4 to have higher percentages of significant alternative-acceptor (Supplementary Fig. S28a; L1–3 95%-confidence interval [1.86, 3.21]; L4 95%-confidence interval

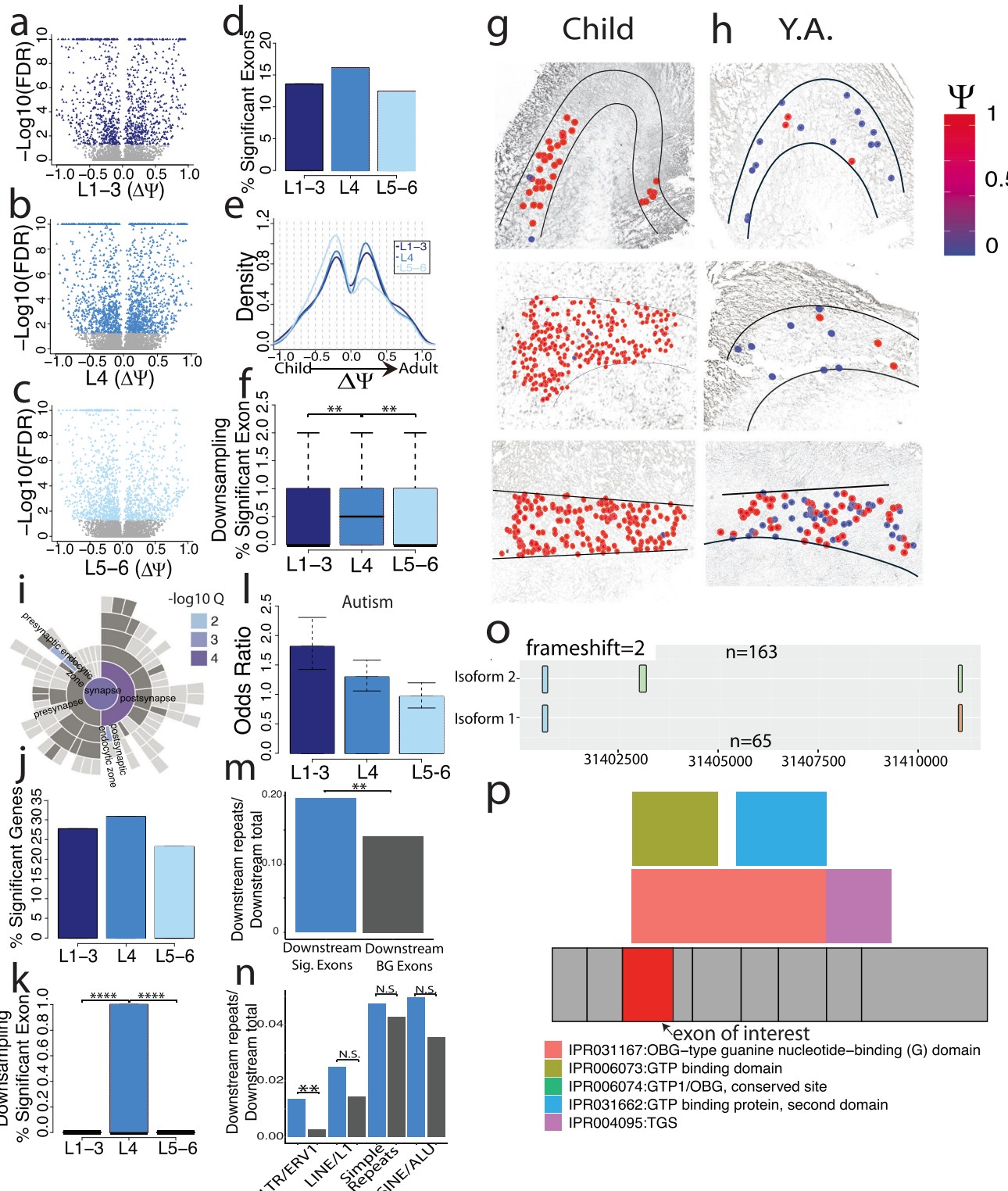

[3.46, 5.12]; L5–6 95%-confidence interval [1.68, 2.91]) as well as poly(A)-site regulation in the considered developmental period (Fig. 6j and Supplementary Fig. S29). However, we note that these values were not downsampled due to a lack of sufficient data amounts. For alternative donors, we found a visually similar trend, however with overlapping 95%-confidence intervals (Supplementary Fig. S28b; L1–3 95%-confidence interval [2.36, 3.93]; L4 95%-confidence interval [3.52, 5.31]; L5–6 95%-confidence interval [2.86, 4.47]). Layer 4 similarly showed increased changes compared to Layer 5–6 in full-length isoform usage

(Supplementary Fig. S26c). We had the largest amount of overall read numbers for excitatory neurons, which enabled downsampling experiments using exclusively spots that are representing excitatory neurons only. These downsampling experiments revealed that the higher ratio of splicing regulation in layer 4 is confirmed when only considering excitatory neurons (L1–3: Mean = 0; SD = 0; L4: Mean = 0.26; SD = 0.44; L5–6: Mean = 0; SD = 0; corrected 2-sided Wilcoxon rank sum test; L1–3 v L4 FDR = $5.0 \times 10^{-8}$; L4 V L5–6 FDR = $5.0 \times 10^{-8}$; Fig. 6k). We then compared the lists of genes with altered alternative

**Fig. 6 | Layer 4 shows the strongest splicing alterations among cortical layers. a** Exons tested between ages in L1–3 by deltaPSI (Y.A.-Child) and -log10(FDR). Colored points are significant exons and grey points are non-significant. **b** Exons tested between ages in L4 by deltaPSI and -log10(FDR). Colored points are significant exons and grey points are non-significant. **c** Exons tested between ages in L5–6 by deltaPSI and -log10(FDR). Colored points are significant exons and grey points are non-significant. **d** Percent significant of tested exons and |deltaPSI| ≥ 0.2 for L1–3, L4, and L5–6. L1–3 percent significant: 13.57%; L4 percent significant: 16.11%; L5–6 percent significant: 12.44%. **e** deltaPSI density of significant exons for L1–3, L4, and L5–6. **f** Percent significant exons from downsampling experiments. Exon selection and percent significant calculations are repeated 100× and plotted ("Methods", *n* = 100). Boxplot center line is at the median, upper bound at the 75th percentile, lower bound at the 25th percentile with whiskers at 1.5 × IQR. **g** Exon chr12_85027899_85028023_– in the *TSPAN19* gene plotted by area in 3 child samples. Color indicates exon inclusion or PSI. **h** Exon chr12_85027899- 85028023 in the *TSPAN19* gene plotted by area in 3 Y.A. samples. **i** SynGO location enrichment of significant L4 genes. **j** Percent significant genes with alternative poly(A) sites by layer. L1–3 percent significant = 17.18%, L4 percent significant = 21.32%, L5–6 percent significant = 15.50%. **k** Downsampling of excitatory neuron-specific reads and genes. **l** Odds ratio comparing significant group v background group of autism associated genes. L1–3 95%-confidence interval [1.42, 2.30], *n* = 3101; L4 95%-confidence interval [1.06, 1.58], *n* = 4154; L5–6 95%-confidence interval [0.77, 1.20], *n* = 4745. **m** Ratio of sequences with repetitive elements found to total number of sequences per group. Sig: Sequences found upstream of exons with |deltaPSI| > 0.5 and FDR < 0.05. BG: Sequences found upstream of exons with |deltaPSI| <0.05 and FDR > 0.05. **n** Ratio of sequences with repetitive elements found to total number of sequences per group, broken down by repetitive element. **o** RiboSplitter plotted example of an exon skipping event in the DRG1 gene. Blue squares indicate constitutive exons, green square indicate alternative exon. "*n*" indicates number of reads in each isoform. **p** DRG1 gene plotted where exons are denoted in grey squares. Colored boxes indicate protein domains and their locations relative to exons. **** = *P* ≤ 0.0001, *** = *P* ≤ 0.001, ** = *P* ≤ 0.01, * = *P* ≤ 0.01. A two-sided Wilcoxon rank-sum test was applied to all the comparisons shown in (**f** and **k**). A corrected 2-sided Fisher test was applied to (**m** and **n**).

splicing during puberty to published lists with disease-associated splicing. For a published list of genes with autism-spectrum disorder-associated splicing changes[12,65,66], we found a higher odds-ratio for L1–3 compared to L5–6, with non-overlapping 95% confidence intervals. The L4 odds ratio was in between and had overlapping 95% confidence intervals with both L1–3 and L5–6 (L1–3 95%-confidence interval [1.42, 2.30]); L4 95%-confidence interval [1.06, 1.58]; L5–6 95%-confidence interval [0.77, 1.20]; Fig. 6l). Similarly investigating gene lists of amyotrophic lateral sclerosis (ALS)[67], schizophrenia[68] and Alzheimer's disease (AD)[11], we found non-significant associations for all diseases and layers, when correcting for the three tests performed per disease (Supplementary Fig. S30). We next examined if any repetitive elements existed either upstream or downstream of developmentally-regulated exons. We found that while there was no increased abundance of repetitive elements upstream (2-sided Fisher test: *p* = 0.334; Supplementary Fig. S31), the regulated exons had significantly more repetitive elements downstream compared to a background set (2-sided Fisher test: *p* = 0.004; Fig. 6m; "Methods"). When investigating if this effect was driven by one or multiple types of repetitive elements, we found that LTR/ERV1 repeats, out of the most abundant elements, were increased downstream of developmentally-regulated exons (corrected 2-sided Fisher test: FDR = 0.028). The three other types of repeats we monitored did not reach significance on their own but suggested a similar trend (Fig. 6n).

Additionally, RiboSplitter[69] was used to identify protein domains associated with spliced L4 genes. RiboSplitter requires another alternative splicing analysis software, Spladder[70], to identify alternative splicing events and their related protein domains. Spladder identified 94 exon-skipping events across development in Layer 4. Among these 94 exons, we considered those that are also tested in our data for correlation analysis. $\Delta\Psi$ s from the two sources correlate highly (Spearman's R = 0.83, Supplementary Fig. S32a). In addition to the 94 exon skipping events, RiboSplitter also identified alternatively included retained introns, 3′, 5′, mutually exclusive exons, and multiple skipped exons between age groups (Supplementary Fig. S32b). Of these, 47.2% led to a frameshift mutation. The PDZ domain (IPR001478) was most affected by exon skipping and no domain was most prominently affected by intron retention (Supplementary Data 6). Of note, PDZ domains are often found in scaffolding proteins, which help to organize and shape the postsynaptic landscape[71–73]. An example of an alternative exon which overlaps with a protein domain can be seen in the Developmentally Regulated GTP Binding Protein 1 (*DRG1*) gene. In this example, the highlighted exon is 59% more included the young adult age group compared to the child group, where a total of 163 reads contain the exon and 65 skip it (Fig. 6o). Of note, the removal of this exon changes the reading frame of this protein, and the exon itself spans several protein domains, including OBG-

type guanine nucleotide-binding (G) domain and GTP binding domain (Fig. 6p). The *DRG1* gene has been implicated with microtubule dynamics[74], ribosome regulation[75,76], as well as several developmental neurological disorders including autism spectrum disorder[77] and microcephaly[78].

## Discussion

Developmental changes happening around puberty are known to affect cortical plasticity. However, splicing and poly(A)-site changes in specific layers and cell types, and if these spatially defined changes contribute to neuronal plasticity equally, had not been mapped out completely. Here, we develop a new technology of spatial isoform sequencing at near-single-cell resolution (Spl-ISO-Seq) as well as a software suite (Spl-IsoQuant) enabling barcode deconvolution and isoform expression analysis. In general, the ability to capture single cells with most spatial technology may be limited to the size of the cells within the tissue being studied. Here, we use human brain tissue, which contains an increase in neuronal cell size and density compared to other mammals, where neuronal soma volume can be greater than 2000 um³ [79–81]. The literature gives abundant examples of alternative splicing influencing synaptic function, transmission, and plasticity[82–86]. However, which splicing events are altered in puberty and in which cell types and which cortical layers is much less studied. Armed with the above technological advances, we find that the cortical layers overall show higher developmental splicing changes than the white matter and that these splicing changes harbor enrichment for different types of protein domains. We also noticed that cortical genes containing developmentally regulated exons are enriched for developmentally regulated poly(A)-site usage as well. However, for oligodendrocytes, we find the opposite trend: Higher splicing regulation occurs in the white matter than in the cortex. This raises the compelling question of which variable may be influencing oligodendrocyte splicing in this region-specific way, which could include microenvironmental factors or unique cell-type interactions.

Comparing distinct layers, L4 stands out in developmental regulation of alternative-exon inclusion, alternative-acceptor usage, and Poly(A)-site usage. Results for alternative donors suggest a similar observation, which is however, non-significant. This emphasizes the need for the exciting future direction of monitoring all RNA variables, including splice sites, Poly(A) sites, TSS sites, and RNA modifications during this developmental period. For alternative exons, the importance of L4 in developmental regulation can be traced to excitatory neurons—the most abundant data source in this spatial experiment. We also find in layer 4 that the regions downstream of developmentally regulated exons are enriched with repetitive elements, an observation which is driven by LTR/ERV1 repeats. Literature suggests that these repeats could be influencing splicing themselves, or perhaps act as a

binding site to RNA-binding proteins or other regulatory elements, as it's known that many RNA-binding proteins bind to repeats[87–91]. Interestingly, these repetitive elements have been shown to be dynamically expressed throughout development in the brain[92,93]. Overall, these findings show that LTR/ERV1s co-localize with developmentally-regulated exons. This observation suggests the exciting hypothesis that these repeats may play an important role in alternative splicing in the context of human cortical development, suggesting multiple future research directions. Importantly, L4 is distinct from other cortical layers as it receives signals from the thalamus, whereas other layers generally assist in further sensory processing. Additionally, L4 has a unique structure consisting of ocular dominance columns. Our results align with previous research demonstrating that L4 in the visual cortex exhibits increased synaptic plasticity compared to other layers[94–96] especially during the critical period of sensory development[97–99]. Overall, we find that the splicing changes detected during puberty are linked to synaptic and especially post-synaptic biology as well as autism spectrum disorder. For this disorder L1–3 have stronger association than L5–6, with L4 showing an in-between result, suggesting layer-specific effects of splicing in ASD. We do not find similar associations to AD and ALS—which could be due to these diseases being associated to older ages or different cortical regions and not the period or anatomical area we investigate here—or because of differences in how distinct prior publications derived these gene lists. Of note, we also do not find an association to schizophrenia. A biological explanation may be that schizophrenia arises usually after the age of 20, which is past the age-range we investigated here, while autism arises earlier in childhood. However, technical differences in how the schizophrenia gene list was derived cannot be ruled out. Of note, the autism association of isoform usage is at this point not yet rooted in a specific cell type. The literature suggests that autism is linked to an imbalance of excitation and inhibition, perhaps driven by inhibitory neuron dysfunction[100–102]. Our data is dominated by excitatory neurons—with little conclusions to be drawn about inhibitory neurons. There are two possible, non-mutually exclusive, explanations for our observations. First, inhibitory neuron dysfunction could affect excitatory neurons and their splicing program due to changes in activation patterns. Second, isoforms could be altered in similar ways in both excitatory and inhibitory neurons, but have stronger functional effects in inhibitory neurons.

At the moment, this technology requires sufficient amounts of data between comparisons, which limits the ability to investigate intra-layer variability, especially at the cell-type level or even between two individual cells. Enhancing the rate of assigned singlets and UMIs per barcode captured will enable this future analysis and greatly inform the fields of neuroscience and spatial transcriptomics. Of note, cell-type information was determined with short-read sequencing of fragmented cDNAs due to enhanced read depth and gene expression profiles using short reads. Long-read sequencing (here of unfragmented cDNAs), although necessary to describe transcript information spanning multiple exons, generally is not sufficient for gene-expression based analyses (unless multiple flow cells are used to match short-read depth per sample). Vice versa, short-read sequencing is not sufficient to identify alternative exon inclusion nor characterize fully synthesized molecules. Thus, both sequencing modalities are required in tandem for Spl-ISO-seq.

Additionally, while this technology increases the average length and transcript coverage per read compared to the standard, most reads are not full-length. This drawback limits the extent to which we can describe 5′ TSSs and full-length isoform changes across development. Other technologies, including but not limited to 10× Genomic's Visium, DBiT-Seq, and StereoSeq, use a transcript switch oligo (TSO) rather than a universal priming site (UPS) during 2nd strand synthesis, which would enable full-length molecule capture. However, other variables, including spatial resolution (spot size), molecule diffusion, and capture efficiency, should also be considered when choosing a spatial transcriptomic technology. Future work should focus on optimizing high-resolution and low diffusion spatial technology for mapping not only of full-length isoforms across a region, but also other modalities at the same time. This would propel the field forward and enable the investigation of multi-modal networks in relation to their spatial location and microenvironments.

Overall, we demonstrate a near-single cell spatial transcriptomic technology which maximizes transcript coverage and employ it to better understand alternative splicing in the developing visual cortex. Our new technologies enable a view of splicing changes that can at once be rooted in specific cell types as well as specific areas or structures within anatomical regions. Thus, there are wide applications for this technology in cancer, neurodegeneration, neuropsychiatric disorders, and many other diseases.

# Methods

## Human brain tissue acquisition

The fresh frozen human brain tissues of 4 children (ages 8–11) and 4 young adults (ages 16–19) were obtained from the NIH Neurobiobank at the University of Maryland, MD. Human tissue samples were compliant with research ethics stated by the NIH. All donors completed University of Maryland IRB-approved consent documents or State of Maryland Department of Health IRB-approved consent documents. They were informed via these consent documents that donated tissue would be used for distribution to qualified researchers and that such distributions could be made at any time in the future. These consent documents also assured that the identity of the donor would remain unknown to any tissue recipients and those reviewing the results of their work. As the data generated with the tissue from these donors is limited to non-identifying RNAseq data, it does not lend itself to identification of the donors and is therefore suitable for publication in print or in digital form in publicly accessible databases.

## Spatial barcoding and cDNA synthesis

Tissue was embedded in Tissue-Tek OCT Compound (Sakura, catalog no. 4583) and stored in −80 °C overnight. Tissue blocks and CryoCube (Curio Bioscience part no. JW001) were placed in a −20 °C Leica CM3050 S cryostat for 20 min prior to sectioning. Tissue was sliced at 10 µM thickness. Tissue was positioned and melted over a Curio Seeker 3 × 3 mm slide (Curio Bioscience part no. SK004). CryoCube was sliced at 30 µM thickness and positioned and melted on top of tissue on seeker slide. Tissue hybridization, reverse transcription, tissue clearing and bead resuspension, 2nd strand synthesis, and cDNA amplification were performed according to the Curio Seeker 3 × 3 spatial mapping kit v 2.1. Glass slides with tissue were immediately placed into 1.5 mL tubes with 200 µL hybridization mix (190 µL hybridization buffer [Curio Bioscience part no. B005] & 10 µL RNAse Inhibitor [Curio Bioscience part no. E001]) for 15 min at room temperature. Following hybridization, slides were dipped into RT wash buffer (40 µL RT/SS buffer [Curio Bioscience part no B006] & 160 µL nuclease-free water [Thermofisher catalog no. AAM9937]) for 3 s and transferred to tubes with 200 µL RT reaction mix (40 µL RT/SS buffer, 20 µL dNTPs [Curio Bioscience part no. N001], 5 µL RNase inhibitor, 10 µL RT Enzyme [Curio Bioscience part no. N001 E003], 125 µL Nuclease-free water) for 10 min at room temperature followed by 30 min incubation at 52 °C. 200 µL of Tissue clearing reaction mix (196 µL TC buffer[Curio Bioscience part no. B004], 4 µL TC Enzyme[Curio Bioscience part no. E002]) was added to tubes and incubated for 30 min at 37 °C. 200 µL bead wash buffer (Curio Bioscience part no. B003) was added to tubes and barcoded beads were dissociated from the glass slide by gently pipetting the mix over the area. Once beads were fully dissociated, the glass slide was removed from the tube. Beads were washed by centrifuging for 3000 g for 2 min, supernatant removed, and resuspended in 200 µL bead wash buffer. This washing cycle was repeated once

more prior to second strand synthesis. Beads were then incubated at 95 °C for 5 min and washed. 200 µL second strand mix (40 µL RT/SS buffer, 20 µL dNTP, 2 µL SS Primer [Curio Bioscience part no. P002], 5 µL SS Enzyme [Curio Bioscience part no. E007], 133 µL Nuclease-free water) was added and incubated at 37 °C for 1 h. Beads were then washed 2× and resuspended in cDNA amplification mix (100 µL cDNA amp buffer [Curio Bioscience part no. B007], 8 µL cDNA amp primer mix [Curio Bioscience part no. P003], 4 µL cDNA amp enzyme [Curio Bioscience part no. E006], 88 µL Nuclease-free water). A PCR with the bead mix was conducted as follows: 98 °C for 2 min, followed by 4 cycles of 98 °C for 20 s, 65 °C for 45 s, and 72 °C for 3 min, an additional 14 cycles of 98 °C for 20 s, 67 °C for 20 s, 72 °C for 3 min, followed by 72 °C for 5 min. cDNA was cleaned up with 0.6× SPRIselect beads, incubated for 5 min at room temperature, and washed 2× with 80% ethanol. SPRIselect beads were eluted in 50 µL nuclease-free water and cleaned up with another round of 0.6× SPRIselect beads, ethanol washes, and elution. cDNA concentration was measured with Qubit and cDNA length was measured with Tapestation.

## Tagmentation and illumina library preparation

5 ng of cDNA from the Curio seeker protocol was eluted to 5 µl with nuclease-free water. Illumina libraries were prepared using the Nextera XT Library Kit (Illumina, part no. FC-131-1024) and followed protocol as provided from Curio Biosciences. 10 µL of TD and 5 µL ATM were added to the tube and incubated at 55 °C for 5 min. Immediately after 5 µL of NT was added to the mix and incubated at room temperature for 5 min. 15 µl of NPM and 5 µL of a unique F and R primer pair were added per sample. A PCR with the mix conducted as follows: 72 °C for 3 min, 95 °C for 30 s, followed by 12 cycles of 95 °C for 10 s, 55 °C for 30 s, and 72 °C for 30 s, followed by 72 °C for 5 min. Library was cleaned up with 0.6× SPRI beads, washed 2× with 80% ethanol, and eluted in 50 µl TE. 0.8× SPRI beads were added to the elution and washed with ethanol 2× and eluted in nuclease-free water. cDNA concentration was measured with Qubit and cDNA length was measured with a tapestation. Nextera XT libraries were sequenced with the Illumina Nova Seq X Plus.

## Exome enrichment of spatial cDNA

The following steps were performed during exome capture. Exome capture was performed as previously described to selectively amplify spliced molecules[36]. Agilent probe kit SSELXT Human All Exon V8 (Agilent, 5191–6879) and the reagent kit SureSelectXT HSQ (Agilent, G9611A) were used according to the manufacturer's manual. First, the block oligonucleotide mix was made by combining 1 µl of PCR handle primer (5′-CTACACGACGCTCTTCCGATCT-3′) and 1 µl of UPS primer (5′-AAGCAGTGGTATCAACGCAGAGT-3′) at 200 ng/µl. Next, 5 µl of 50–100 ng/µl spatial untagmented cDNA diluted from the first step was diluted with 2 µl blockmix and 2 µl nuclease-free water (NEB, AM9937). The mix was incubated on a thermocycler under the following conditions: 95 °C for 5 min, 65 °C for 5 min and 65 °C on hold. Next, hybridization mix was made by combining 20 µl of SureSelect Hyb1, 0.8 µl of SureSelect Hyb2, 8.0 µl of SureSelect Hyb3 and 10.4 µl of SureSelect Hyb4 at room temperature. Once the cDNA block mix reached to 65 °C on hold, 5 µl of probe mix, 1.5 µl of nuclease-free water, 0.5 µl of 1:4 diluted RNase Block and 13 µl of the hybridization mix were added to the cDNA block oligo mix and incubated for 24 h at 65 °C. M-270 Streptavidin Dynabeads (Thermo Fisher Scientific, 65305) were prepared by washing three times and resuspended with 200 µl of binding buffer. Once the incubation period ended, cDNA block oligo mix was let sit at room temperature and thenfol mixed with M-270 Dynabeads. The bead-cDNA mix was placed on a Hula mixer for 30 min at room temperature. During the incubation, 600 µl of wash buffer 2 (WB2) was transferred to three wells (200 µl each) of a 0.2-ml PCR tube and incubated in a thermocycler on hold at 65 °C. After the 30-min incubation, the bead buffer was replaced with 200 µl of wash buffer 1

(WB1). Then, the bead-cDNA mix was put back into the Hula mixer for another 15-min incubation with low speed. Next, the WB1 was replaced with WB2, and the tube was transferred to the thermocycler for the next round of incubation (10 min). WB2 was replaced 2 more times at 10 min intervals with the pre-heated WB2. Following incubation, beads were resuspended in 18 µl nuclease-free H2O and then the spliced cDNA, which is attached to the beads, was amplified as described below.

## Long-cDNA capture and amplification

The spliced cDNA captured by exome probes, which bound with the M-270 Dynabeads, was amplified with primers (UPS: 5′-AAGCAGTGG-TATCAACGCAGAGT-3′; PCR: 5′-CTACACGACGCTCTTCCGATCT-3′) and KAPA-HiFi enzyme by using the following PCR protocol: 95 °C for 3 min, 8 cycles of 98 °C for 20 s 64 °C for 60 s and 72 °C for 3 min. The amplified probe captured cDNA was isolated from M-270 beads as supernatant and followed by a size selection/purification with 0.48× SPRIselect beads (in 1.25 M NaCl-20% PEG buffer) and then eluted in 25 µl EB buffer. All 25 µl size-selected spliced cDNA was used as template for the second round amplification of 6–8 cycles with the same PCR conditions suggested above. The product of the second round PCR was size selected/purified with 0.48× SPRIselect beads (in 1.25 M NaCl-20% PEG buffer) and then eluted in 50 µl EB buffer.

## Oxford nanopore library preparation

For all samples, ~75 fmol exome-enriched and length-optimized cDNA underwent ONT library construction by using the Ligation Sequencing Kit (ONT, SQK-LSK114), according to the manufacturer's protocol (Nanopore Protocol, Amplicons by Ligation, version ACDE_9163_v114_revO_29Nov2022). The ONT library was loaded onto a PromethION sequencer by using PromethION Flow Cell (ONT, FLO-PR114M) and sequenced for 72 h. Base-calling was performed with Super-high accuracy settings on the Oxford Nanopore MinKNOWUI dorado basecaller.

## Hematoxylin and Eosin staining

The 10 uM tissue following the experimental tissue slice for each sample was placed on a colorfrost plus slide (Epredia catalog no. 6776214) and fixed in 4% PFA (Electron Microscopy Sciences catalog no. 15710) for 30 min on ice. Slides were washed in 1× PBS 3× for 5 min each. H&E staining kit (abcam catalog no. ab245880) was used to stain slides. Incubation times for Hematoxylin and Eosin were optimized per sample using excess tissue. Slides were imaged on a Leica CTR 6500.

## Short read data processing and deconvolution

Illumina data was analyzed using the Curio Seeker Bioinformatics pipeline (https://curiobioscience.com/bioinformatics-pipeline/). Resulting.rds files were inputted into Robust Cell Type Decomposition (V2.2.1) program[41] with a reference file of including 7 human single nuclei PFC samples with annotated clusters/celltypes, including excitatory neurons, inhibitory neurons, oligodendrocytes, oligodendrocyte precursor cells, astrocytes, microglia, vascular endothelial cells, T cells, fibroblasts, and unknown. Basic RCTD parameters were used in "doublet" mode following published tutorials (https://github.com/dmcable/spacexr/blob/master/vignettes/spatial-transcriptomics.Rmd). Spots designated as "singlets" were assigned as cell types per sample.

## Identifying cortical layers

Cortical layers were identified by data from combining H&E images, excitatory neuron density based on RCTD singlets, and known gene expression markers, which are layer-specific or white matter-specific. These were assigned on a slide-by-slide basis using established visual cortex architecture as a guide.

## Detecting differential expressed genes

For each age group and cell type, the set of differential expressed genes (DEG) detected from the comparison was obtained by running FindMarkers function of Seurat[103]. The GO enrichment analysis for the DEGs was performed by using enrichGO or enrichKEGG function of clusterProfiler[104] 4.2.2.

## Long read data processing

**Spl-IsoQuant overview.** To process spatial long-read sequencing data, generated from the previously mentioned methods, we developed a novel pipeline, Spl-IsoQuant, which is based on IsoQuant—the software for bulk long-read data analysis previously developed in the lab[49]. Spl-IsoQuant starts with barcode calling to identify barcode and UMI sequences in every read. Further, reads are mapped to the reference genome with minimap2 in spliced mode[105] and assigned to known genes and isoforms using existing IsoQuant algorithms. Further, PCR duplicates are removed using the obtained gene assignments and identified UMI sequences. The resulting collection of reads represent unique known mRNA molecules and their respective cells, which is further used for analysis of polyA sites, TSS, and splicing patterns.

Spl-IsoQuant is an open source software and is openly available at https://github.com/algbio/spl-IsoQuant.

**Barcode detection.** To identify linker, primer, and barcodes positions in a sequenced read, we implement a pattern-matching algorithm based on short k-mer indexing and conventional Smith–Waterman local alignment. Given a set of known sequences (e.g., barcode whitelist, a linker, or a primer), a k-mer index is a key-value hash table containing all possible k-mers as keys and a set of known sequences containing this k-mer as a respective value. Thus, to quickly detect a presence of a known sequence in the read, one needs to split the read (or its region) into k-mers and extract their corresponding values for the index. We construct 3 distinct k-mer indices: for the primer sequence ($k = 6$), the linker ($k = 5$) and the last one for the list of all known barcodes ($k = 6$). Small $k$ values are used to be able to detect exact k-mer matches even when the read has sequencing errors. However, short k-mer matches can be accidental, especially for a barcode whitelist containing tens of thousands of barcodes. Thus, a k-mer index can only be used to obtain a set of preliminary candidate sequences. Further, we use the classic Smith–Waterman (ssw-py library[106]) local alignment algorithm to check which k-mer matching derived candidates are actually present in the read and filter out arbitrary matches. Exploiting the k-mer index approach allows to shrink the search region when matching the primer and the linker, as well as greatly reduce the number of Smith–Waterman calls when detecting barcode sequence.

To detect barcodes and UMIs, reads are processed individually using the following procedure. First, a poly-T stretch is detected using a sliding window technique. Further, the primer and the linker positions are identified as described above in the region of the read preceding the polyT stretch. Primer and linker coordinates are used to extract a potential barcode sequence (Fig. 3b). This sequence is further split into k-mers and possible matching barcodes from the whitelist are detected using the barcode k-mer index. Among those, the candidate with the highest Smith–Waterman local alignment score is selected as the barcode for this read. If the best alignment score is below a certain threshold, the candidate barcode is considered to be unreliable and is not reported. We exploit the classic local alignment scoring system that adds 1 for a symbol match and −1 for mismatch or indel, the default alignment score cutoff is 13. Although this scheme does not allow errors in the middle of the barcode, it does consider the possibility of terminal indels/substitutions, including ones caused by potential inaccuracy in barcode positioning. UMI is simply extracted from the read between barcode end and polyT start. If no polyT stretch, linker, primer, or barcode are found, the above procedure is

repeated for the reverse-complemented read sequence. If no reliable barcode is detected for both forward and reverse sequence, no barcode and UMI are reported and the read is ignored in the further analysis.

The described algorithm is implemented in Python with the use of the SSW library[106] and is as a part of the Spl-IsoQuant pipeline. An ONT dataset containing 85 million reads is processed in approximately 2 h using 20 threads. In contrast, mapping the same dataset with minimap2 takes about 18 h, and the remaining part of the pipeline (read-to-isoform assignment and UMI deduplication) takes roughly 4 h on the same dataset.Read processing and UMI deduplicationOnce barcodes are detected, the reads are mapped to the reference genome using minimap2 with the default setting used in IsoQuant (minimap2 -x splice –junc-bed annotation.bed -a −MD -k 14 -Y –secondary = yes). Spl-IsoQuant then assigns all mapped reads to the reference genes and isoform using the algorithm described in (Prjibelski et al., 2024). Multimapped reads are resolved by selecting the alignment that is the most consistent with the reference annotation or discarded. Reads that were uniquely assigned to a known gene are further processed to remove PCR duplicates.To filter out PCR duplicates, reads are grouped by the assigned gene and barcode. Thus, reads in the same group represent mRNAs originating from the same cell and the same gene. If a pair of reads in the group contain identical UMI sequences, it is highly likely they were sequenced from PCR duplicates of the same mRNA molecule. However, due to the presence of sequencing errors, we consider reads to be PCR duplicates if their UMI sequences have a certain level of similarity, namely, they are edit distance 2 apart or less (starting and trailing indels are not considered). To identify PCR duplicates within a set of reads having the same barcode and the same assigned gene, we iterate over these reads one by one and collect a set of "representative" UMIs (empty at the beginning of iteration). If a read contains a UMI that is similar to any of the collected representative UMIs, this read is assigned to the most similar representative UMI. Otherwise, the UMI is added to the representative set and the read is assigned to its own UMI. Although this algorithm might not be exact and requires quadratic number of UMI comparisons in the worst case, in practice, it allows avoiding computing all-vs-all pairwise edit distances.Further, among reads assigned to the same representative UMI, we select the one covering the highest number of splice junctions. Other reads assigned to this representative UMI are deemed to be PCR duplicates and thus are further ignored. UMI-filtered reads are used in the downstream analysis, such as exon and transcript quantification and differential splicing analysis.Indeed, it is possible that similar UMIs were attached to two distinct mRNA molecules and the described procedure will remove a read which does not have a PCR duplicate. However, due to randomness of UMI sequences (~150 K distinct possible UMIs of length 9 bp with last to non-T bases), such an event is rare and is unlikely to affect the downstream analysis.

**Benchmarking spl-IsoQuant.** To assess the designed barcode calling algorithm, we simulated Nanopore data with Trans-NanoSim[107]. We used a forked version of Trans-NanoSim (https://github.com/andrewprzh/lrgasp-simulation) with the improved read truncation procedure, which allowed us to simulate data with realistic ONT truncation probabilities, as well as reads with no sequence truncation. This version of Trans-NanoSim was previously used to benchmark IsoQuant[5]. The NanoSim error model was trained using actual sequencing data produced in this project (sample 329), exhibiting a sequencing error rate of 4.4% (1.4% mismatches, 2.0% insertions, 1.0% deletions). Therefore, recall and precision estimated with simulated data accurately represents a real-life situation.

Molecule templates provided to Trans-NanoSim were generated by concatenating a reference transcript sequence with 30 bp poly(T) stretch, a random UMI, an arbitrary barcode from the whitelist, the linker and the primer according to the molecule scheme (Fig. 2h).

Reference transcripts were randomly selected according to the expression profile that was also derived from the same sample (329). Five million distinct molecule templates were generated, featuring ~70,000 distinct barcodes. In total, 20 M Nanopore reads were simulated for the benchmarking.

We Spl-IsoQuant ran on the simulated dataset and detected barcodes were compared against true barcodes attached to mRNA sequence during the simulation. In this procedure, correctly called barcode is regarded as true positive, while wrongly reported barcode— as false positive. Reads, for which no barcode is detected due to, for example, high number sequencing errors or read truncation, are considered as false negatives.

**Calling differentially included exons.** After Spl-IsoQuant was run on the long read data and identified reads mapped, barcoded, and spliced reads with corrected UMIs, an AllInfo file was generated containing the following information: Read name, Gene, Group for comparison (ie. cell type or region), barcode, UMI, intron chain, TSS site, PolyA site, exon chain, known or novel transcript, and number of introns. More information regarding AllInfo files can be found at https://github.com/noush-joglekar/scisorseqr[27]. We separated AllInfo files by which comparisons were being performed. For example, if the comparison of child cortex vs. young adult cortex was being performed, reads which aligned with each comparison were identified and placed into 2 distinct AllInfo subfiles. These subfiles were run in the casesVcontrols() function of the scisorATAC R package with minimum reads required per exon set to 10, in order to identify which exons are differentially included between the 2 groups using basic parameters.

First, we identify alternative exons using reads from both groups that are being compared (e.g., layer 4 barcodes from young individuals and layer 4 barcodes from old individuals).

This uses only reads that

– Have introns
– Respect the splice site consensus at all of their introns

We identify exons as alternative "cassettes" using the following approach:

For each exon, we collect the following numbers

– A: Number of reads that support the exon with both splice sites
– B: Number of reads that support the left splice site with the right end of the read ending on the exon
– C: Number of reads that support the right splice site with the left end of the read ending on the exon
– D: Number of reads skipping the exon (but being assigned to the same gene)
– E: Number of reads overlapping the exon
– First, we exclude exons, for which $A + B + C + D < 10$. For these exons there are simply too few reads to perform any cluster-specific analysis.

Second, we exclude exons, for which $(A + B + C + D)/(A + B + C + D + E) < 0.8$. These are exons, for which less than 80% of overlapping reads clearly support inclusion or exclusion of the exon. In these cases a binary view of "inclusion vs. exclusion" is not advisable.

Third, we calculate a PSI for the left part of the exon ("leftPSI") as $(A + B)/(A + B + D)$ and a "rightPSI" as $(A + C)/(A + C + D)$. We keep the exon if and only if both leftPSI and rightPSI are contained in the interval [0.05, 0.95].

For the exons remaining after this step, we proceed to calculating cluster-specific (i.e., layer 4 in "old" individuals) exon inclusion and skipping reads as well as a cluster-specific PSI as $(A + B + C)/(A + B + C + D)$ if and only if there $A + B + C + D >= 10$. Exclusion and inclusion reads from the two groups (i.e., layer 4 "old" and layer 4 "young") are then used to populate a $2 \times 2$ table. A table is kept for statistical testing if and only if three of the four cells in the $2 \times 2$ table have expected counts of 5 or higher.

The remaining exons (represented each by one table) are testing for a significant association of cell type and exon inclusion using two-sided Fisher test. Calculated p-values are then corrected for multiple testing using the Benjamini-Yekutieli correction.

**Calling differentially included isoforms.** After files containing all barcoded and UMI corrected reads with corresponding gene and transcript information (AllInfo files) were generated as described above, files were run through the IsoQuant() and DiffSplicingAnalysis() functions from the ScisorSeqR package (Joglekar et al., Nature Communications, 2021). The IsoQuant function quantifies unique CAGE and PolyA sites, as well as whole isoforms, per gene and barcode. The DiffSplicingAnalysis() function identifies differential isoform usage by spatial region and age comparisons. Each isoform is assigned an ID based on the TSS, introns, and polyA-site information, with isoform 1 being most abundant per gene. Isoform counts were then identified by which comparison they originated from (i.e., child v. young adult). Genes were filtered out if they did not reach the minNumReads criteria (set to 10). Remaining genes were tested in an maximum of $11 \times 2$ matrix of counts, where a maximum of 10 isoforms were identified, with all remaining isoform counts summed in row 11. P-values were calculated with a chi-squared test and a Benjamini–Hochberg correction. Delta Pi values per gene were calculated as the sum of change in percent isoform from the top two isoforms detected.

**Calling differentially included PolyA sites.** We determined the PolyAs per read as assigned in the Allinfo file. Files were run through the DiffSplicingAnalysis() function, where typeOfTest = "PolyA", which identifies alternative PolyA sites by group. PolyA sites per gene were assigned an ID, with PolyA 1 being most abundant. Counts were identified per testing condition and summarized in an $11 \times 2$ table as described above. Testing was performed as described above for isoforms.

**Downsampling experiments.** To compare two comparisons (i.e., Cortical differences between age groups with WM differences across age groups) with equal power, we performed downsampling experiments. We first selected all exons which have at least 17 reads coming from 3 or more samples per age group. We then randomly selected 20 reads among the total. These reads were then used to re-calculate the difference in percent isoform inclusion between the ages (ΔPSI). Next, we selected 50 exons randomly for this comparison, enforcing that there be at most exon per gene. We then repeated these steps for all cell types which were compared. This yields 50 $2 \times 2$ tables for all comparisons, with exactly equal column sums and the same characteristics (table number) for correction multiple testing. We then proceeded to carry out Fisher tests and corrections for multiple testing and recorded the number of significant events for all comparisons. The procedure was repeated 100 times giving a distribution of significant percentages for both comparisons. These two distributions were compared with a Wilcoxon rank-sum test. For excitatory neuron sampling in Fig. 6n, we employed cutoffs of 7 reads per sample in at least 3 samples per age group and sampled 15 exons and 30 genes due to smaller sample sizes.

**Identifying last exon length for PolyA differences.** Reads which contained an annotated Poly(A) site were selected. The length of the last exon prior to the Poly(A) tail was calculated based on exon coordinates, respective to the strand direction. The average poly(A) length per gene was calculated per Cortex and WM age group. These lengths were plotted in boxplot format as seen in Fig. 4e. Additionally, we analyzed poly(A) length in genes which have differential poly(A) expression per group. Thus, from the cortical reads which contained

an annotated Poly(A) site, we subsetted those which were mapped to the set of altered poly(A) genes in cortex. We similarly subsetted poly(A) reads from WM, which were mapped to significantly altered genes in WM. We again calculated the average length of Poly(A) exons per gene for each area and age group, which is shown as a boxplot in Fig. 4f.

**Repetitive elements analysis.** The analysis concerning repeat identification and their occurrence was conducted using the RepeatMasker webserver with default parameters (A.F.A. Smit, R. Hubley, & P. Green, unpublished data; Current Version: open-4.0.9 with Dfam 3.0 only). The analysis was carried out on upstream (150-30 upstream bp) and downstream (10–150 downstream bp) regions independently. To evaluate the top 4 types of repeats identified in the signal sequences, their occurrence in a significant group of exons (FDR < 0.05 + |DPSI| > 0.5) was compared to that in background sequences (FDR > 0.05 + |DPSI| < 0.05) using Fisher's exact test. The analysis considered both the individual contributions of each repeat class and their combined effect.

**Getting protein superfamily annotations per exon.** Cortex tested exons were split into a highly variable group (FDR < 0.05 and |DPSI| > 0.5) and a background group (FDR > 0.05 and |DPSI| < 0.1). For each exon per group, we used the genomeToProtein() function of the ensembldb package and extracted the Ensembl ID, coordinates, and residue sequence of the protein identified. We filtered the obtained protein identifiers based on their corresponding Ensembl transcript IDs and limited the search to the principal isoforms from the APPRIS database[108]. For each protein sequence, we ran the SUPERFAMILY[109,110] tool that uses the hidden Markov model to identify the structural-defined SCOP protein domain families and the domain boundaries. The tool was implemented in InterProScan[111–113]. Protein regions not associated with domains were considered interdomain linkers. Only transcripts with domains which spanned exons in background and significant were selected.

**Modeling domain architecture.** Domain architecture of each protein isoform were obtained using structure-driven SUPERFAMILY domain annotation[110] extracted from Ensemble[114]. Structural characterizations of the proteins were done de novo using AlphaFold 3[115] server no structural templates were detected for either protein, preventing a potentially more accurate template-based modeling approach. Some of the long interdomain linkers in the proteins were modeled with low or very low confidence scores (70 > plDDT > 50 and plDDT < 50), correspondingly and were excluded from the final models.

**Ribosplitter protein domain analysis.** Spladder[70] was performed with.bam files with isolated L4 reads per individual and basic inputs as described in the online tutorial (https://github.com/ratschlab/spladder?tab=readme-ov-file), using commands *spladder build* and *spladder test*. The output folder created from *spladder test* was used as the input for RiboSplitter[69], using functions read_details(), read_isoforms() with basic inputs as documented in the online tutorial (https://github.com/R-Najjar/RiboSplitter). Basic tutorial commands were followed in order to generate a table containing differentially expressed events between age groups. Examples were plotted with commands *splicing_figure*() and *domain_fig*() using basic inputs.

**Reporting summary**
Further information on research design is available in the Nature Portfolio Reporting Summary linked to this article.

## Data availability
The data supporting the findings of this study are available from the corresponding authors upon request. The sequencing data generated in this study have been deposited in the National Institute of Health's Sequence Reads Archive (SRA) under accession code PRJNA1116561. Source data are provided with this paper.

## Code availability
The package Spl-Isoquant is available at https://github.com/algbio/spl-IsoQuant[116].

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

## Acknowledgements

We thank Olivia Spicer and the NIH NeuroBioBank for human tissues. We thank Adrian Tan, Chendong Pan, Aihong Liu, Seongeun Oh, and Jenny Xiang from the Genomics Resources Core Facility at Weill Cornell Medicine for performing RNA sequencing. We thank Dr. Christopher Mason for use of his PromethION machine. We also thank Weill Cornell Medicine Scientific Computing Unit (SCU) for use of their computational resources. Supported by: NIGMS 1R01GM135247-01 (H.U.T.), MIRA R35 GM152101-01 (H.U.T.), Brain Initiative grant 1RF1MH121267-01 (H.U.T.), NIDA U01 DA053625-01 (H.U.T. among others), NIDA grant 2T32DA039080 (J.H., N.B.), NSF GRFP # 2139291 (C.F.), the Feil Family Foundation (H.U.T.), European Research Council (ERC) European Union's Horizon 2020 research and innovation programme (grant agreement No. 851093, SAFEBIO) (A.D.P., A.I.T.), Research Council of Finland grants No. 322595, 346968, 358744 (A.D.P., A.I.T.), R01HD111089 (M.E.R.), NLM 1R01LM014017-01 (D.K.), ERC ASTRA_855923 (G.G.T.), National Center for Gene Therapy and Drugs based on RNA Technology (CN00000041), financed by NextGenerationEU PNRR MUR–M4C2–Action 1.4- Call "Potenziamento strutture di ricerca e di campioni nazionali di R&S" (CUP J33C22001130001) (G.G.T., A.V.), MIRA R35 GM138152 (I.H.).

## Author contributions

C.F., W.H., and H.U.T. devised the experiments. C.F., W.H., J.J., B.E., and Y.H. performed the experiments. A.D.P. created and optimized the barcode detection algorithm. A.D.P., C.F., L.M., and H.U.T. devised the analyses. A.D.P., C.F., L.M., A.V., O.N., J.H., N.B., and D.M. performed the analyses. M.E.R., I.H., G.G.T., D.M., A.I.T., and H.U.T. supervised the project.

## Competing interests

H.U.T. has presented at user meetings of 10× Genomics, Oxford Nanopore Technologies, and Pacific Biosciences, which in some cases included payment for travel and accommodations. H.U.T. has recently agreed to consult for ISOgenix Ltd., for work unrelated to the present manuscript. The other authors declare no competing interests.
