## [Transparent Peer Review file · Nature Communications]

A spatial long-read approach at near-single-cell resolution reveals developmental regulation of splicing and poly(A) sites in distinct cortical layers and cell types.

Corresponding Author: Dr Hagen Tilgner

Version 0:

Reviewer comments:

Reviewer #1

(Remarks to the Author)

The authors have thoroughly addressed all of my comments. I particularly appreciated the new simulations and the added technical clarifications. The manuscript has improved significantly, and I fully support its publication in any Nature Press journal.

(Remarks on code availability)

REVIEWER 1

General Statement to Reviewer 1:

Dear Reviewer #1,

We are very grateful for the time and effort, which you put into reading over our manuscript a second time. Your questions and comments have allowed us to clarify and expand on a multitude of topics, gain stronger insights, and modify the manuscript to, hopefully, a more clear and informative form for future readers. These efforts have resulted in 7 new supplementary figures, 1 new supplementary table, and substantial additions to the text's discussion. Overall, we hope the reviewer agrees that this manuscript is now sufficient for acceptance at Nature Communications.

In summary, we were able to (i) emphasize the necessity for short-read sequencing for layer and cell-type identification in several new supplemental figures as well as the discussion, (ii) demonstrate that not only gene expression, but also transcript expression is correlated among datasets, and transcript expression is not strongly influenced by transcript length, (iii) show that the Spl-ISO-seq pipeline does not introduce any significant bias with respect to read length or splice sites, (iv) emphasize that the Long Exome LR sequencing method substantially increases coverage of captured transcripts, and more.

Below we provide point-by-point answers to all comments. Reviewers' questions are written in normal font, the answers are written using **dark blue font** with the new parts of the manuscript highlighted with *cursive*. We also highlight the key summary points in **bold**.

Referees' comments:

Referee #1 (Remarks to the Author):

1) I appreciate the authors response - but A) I would suggest they give more of the details they just gave here in the manuscript and B) I . . . would have thought you could use all the data (even short) if you are doing gene level analysis?

We thank the reviewer for allowing us to clarify this point to the readers. Here we respond to reviewer's points A and B below:

B) First, we respond to point B ("I . . . would have thought you could use all the data (even short) if you are doing gene level analysis?"), in 2 parts. Firstly, **I** we explain that **only the short-read data is used in gene expression analyses**, and secondly, **II** we recognize that the many datasets used across figures may be confusing to the reader and **implement a table to help organize the information**.

B-I) We apologize for not sufficiently explaining which data was used in gene-level analyses. All gene-expression-based analyses (including layer-specific marker identification, gene

ontology (Figures 1F-G), gene-expression heatmap across age and region (Figure 1H)) and marker-gene-expression placement to identify layers were initially conducted **only with Illumina short-read data** (although in response to point A we have now added marker-gene expression with Long Exome long read data). This data is saturated and sufficient to use for such analyses. However, the Long Exome LR data (i.e., exome enrichment followed by long-molecule selection), which is used in figures 2-6, yields a **3.8x lower gene-detection rate** of genes per spot compared to the short-read data (because we actively remove short and/or intronic molecules before long read sequencing). Thus, the long-read data is not sufficient for these analyses, although necessary for detecting spliced molecules.

B-II) We recognize that this manuscript contains many datasets which vary in usage across analyses which may be complicated for the reader to follow. We also apologize for terming the same dataset in several different ways across the manuscript, which only added to the confusion. For example, Figure 2A's "Standard LR" was also called "Control No Exome" in supplemental figures, which we originally thought would clarify the differences we wanted to point out, but in hindsight realize this could be a source of confusion. Below, we again **define the datasets below and have constructed a table illustrating which datasets are used in corresponding main figure panels which we hope clarifies this point**. Additionally, unless indicated in the descriptions below, **have changed all dataset names to be consistent throughout the manuscript**.

Illumina Sequencing:

- **Naive Short Read (SR):** Naive Illumina short-read sequencing of spatially barcoded cDNA from all 8 Curio Bioscience 3x3 mm slide datasets.
- **Spliced Naive SR:** A subset of the Naive Illumina SR data from which only spliced reads are extracted. This data is used in Supplemental Figure S16 and S22. This is a very small subset as most short reads are unspliced.

ONT Sequencing:

- **Standard Long Read (LR):** Naive long-read sequencing of spatially barcoded cDNA from 2 Curio Bioscience 3x3 mm slide datasets. This method is shown in Figure 2, Supplemental Figure S10, Supplemental Figure S11, Supplemental Figure S13, Supplemental Figure S15, and Supplemental Figure S17 to demonstrate method comparison.
- **Standard Exome LR:** Spatially-barcoded cDNA underwent exome enrichment without long-molecule selection (Methods). Enriched cDNA of 2 samples were long-read sequenced. This method is shown in Figure 2, Supplemental Figure S10, Supplemental Figure S11, Supplemental Figure S12, Supplemental Figure S13, Supplemental Figure S14, Supplemental Figure S15, and Supplemental Figure S17 to demonstrate method comparison.
- **Long Exome LR:** Spatially-barcoded cDNA underwent exome enrichment with long-molecule selection (Methods). Enriched cDNA of all 8 samples were long-read sequenced. This dataset is used for all long-read analyses in Figure 4 and following.

Datasets used in each main figure

Figure	Panels	Datasets
Figure 1	B, D-F	Naive Short Read
Figure 2	A-F	Standard LR, Standard Exome LR, Long Exome LR
Figure 4	A-P	Long Exome LR
Figure 5	A-H	Long Exome LR
Figure 6	A-P	Long Exome LR

A) Next, we respond to point A (“I would suggest they give more of the details they just gave here in the manuscript”). We have **added additional explanatory text** to the results and method sections (see below). We have also **added 2 new supplementary figures**, which show increased layer-specific marker expression in short-read compared to long read data (Supplementary Figure S3), as well an example of cell-type deconvolution performed with long-read data which only identified 17% of the previously identified singlets and 5.3% of the previously identified excitatory neurons (Supplementary Figure S4 - see answer 9C for more detailed information). Taken together, these results demonstrate the necessity of the Illumina sequencing in order to increase accuracy of layer and cell-type identification. We have pasted these figures below for convenience.

Modified results section:

“The layers of the visual cortex are distinguishable by key attributes including cell density, layer thickness, and layer-specific markers. Illumina UMI counts per spot suggested the position of distinct cortical layers (Figure 1b), which was confirmed with H&E stains of an adjacent tissue slice (Figure 1c, Supplemental Figure S2), short-read deconvolution methods⁴¹ (Methods), and known layer-specific markers. These layer-specific markers were used to help identify layer cutoffs and were identified with Illumina sequencing, since Illumina was less sparse compared to ONT sequencing (Supplemental Figure S3). We also employed a cell-type deconvolution program which identified which spots entirely contained individual cell types. Most cell-type deconvolution programs require large numbers of reads to accurately deconvolve cell types, thus Illumina sequencing was required for this process as ONT isoform sequencing lacked sufficient depth (Supplemental Figure S4) and could not sufficiently identify layer patterns or cell types (Supplemental Figure S5).”

Supplemental Figure S3

Supplemental Figure S3: **a)** Log₁₀(UMI) counts of Naive Short-Read data per spot of gene CALB1 plotted by spatial location. **b)** Log₁₀(UMI) counts of Long-Exome data per spot of gene CALB1 plotted by spatial location. **c)** Log₁₀(UMI) counts of Naive Short-Read data per spot of gene RORB plotted by spatial location. **d)** Log₁₀(UMI) counts of Long-Exome data per spot of gene RORB plotted by spatial location. **e)** Log₁₀(UMI) counts of Naive Short-Read data per spot of gene MBP plotted by spatial location. **f)** Log₁₀(UMI) counts of Long-Exome data per spot of gene MBP plotted by spatial location. Black lines indicate layer cutoffs.

Supplemental Figure S5

Supplemental Figure S5: **a)** UMIs per spot from Naive short-read data plotted by spatial location. **b)** UMIs per spot from Long-Exome data plotted by spatial location. **c)** Excitatory neuron singlets and their positions identified from inputting Naive Short-Read data into a deconvolution program. **d)** Excitatory neuron singlets and their positions identified from inputting Long-Exome data into a deconvolution program. **e)** Number of identified “Doublet Certain”, “Doublet Uncertain”, “Reject”, or “Singlet” spots by data input. **f)** Number of identified excitatory neuron and oligodendrocyte singlets by data input.

4) If they are not doing strand-switching in Curio - how is this going to work for full-length or knowing the 5'/TSS of the cDNA? Sure, the authors are enriching for longer molecules with their exome capture approach, but isn't the basic premise and the cDNA generation flawed as a result of this?

We thank the reviewer for giving us the opportunity to clarify and reflect on this point. We first again describe (A) how second strands are synthesized and (B) reflect on the reviewer's comment, "but isn't the basic premise and the cDNA generation flawed as a result of this?".

A. The reviewer is correct in their statement that a 5' template switch oligo (TSO) is not employed in any method presented in this paper, which is due to Curio Bioscience's established library preparation protocol where a 5' random hexamer (termed UPS) binds to a random portion of the gene. **Thus, most molecules (> 95%) are not full length as the original library preparation protocol was optimized for short-read sequencing and gene expression analysis only, and was not designed to capture full-length molecules.** In general, cDNA is constructed normally, following the usual steps of reverse transcription, 2nd Strand Synthesis, and cDNA amplification. It is only after the cDNA-amplification step that exome enrichment and long-molecule selection occur. Please see below for the protocol overview diagram which illustrates these steps. We note that this image comes directly from the Curio Bioscience protocol and is not in the manuscript.

B. Here we respond to “but isn’t the basic premise and the cDNA generation flawed as a result of this?” The lack of full-length molecules is due to the nature of the provided Curio Bioscience protocol and kit itself (see A). Here, cDNA is constructed normally, following the usual steps of reverse transcription, 2nd Strand Synthesis, and cDNA amplification. It is only after the cDNA-amplification step that exome enrichment and long-molecule selection occur. Overall, the naive cDNA that is generated first follows the normal steps of the Curio pipeline which aligns with other common cDNA generation protocols, including 10X Genomics. **Thus, naive cDNA generation is not flawed, and only afterwards spliced and long cDNAs are preferentially selected through our exome-enrichment and long-molecule selection protocol.**

Additionally, our long-molecule enrichment **captures substantially more information per read compared to the Standard (naive long read) and Standard Exome LR (exome enriched long read) approaches, increasing the average % of transcript covered** (Supplemental Figure S14, newly added Supplemental Figure S15) **and number of exons captured** (Figure 2c) **without strongly biasing gene or transcript expression** (Figure 2d, newly added Supplemental Figure S12 - see answer 6 for more details). Even though most reads are not full length, we can measure exons further away from the polyA which we couldn't capture with our standard method, of which an example can be seen in Figure 2e. Although we can only reflect on full-length isoforms for a small subset of the data (Supplemental Figure S26), **the increase in information per read is of value to the field, especially for cases where full-length-molecule generation may not be possible.**

Overall, cDNA generation is not flawed, but rather spliced and long cDNAs are selected from the original pool of cDNA. We demonstrate that this selection does not strongly bias gene or transcript expression while also increasing the % transcript covered and # of exons captured. However, we do acknowledge that the method reveals full-length isoforms for only a minority of reads.

5A) I see thier long exome and long-no-exome comparison but . . . that’s just not what’s being reported more recently. On a PromethION or on Revio with Kinnex, typical numbers per flowcell without capture are up in the 75M+ reads - sure that might prior to alignment, and I’ll grant that many of the reads might be less informative, but their explanation of long-primer-dimer complexes and artifacts are . . . not convincing.

We thank the reviewer for this comment. Firstly, we clarify that in the barplot referenced in the original response to reviewers and in 5A (pasted below for convenience), the data being compared is “**Long Exome LR**” (exome enrichment + long molecule selection, which we have referenced before) and “**Long No Exome LR**” (only long molecule selection -- without exome enrichment, which is not discussed in the manuscript).

Barplot referenced in 5A from previous Response to Reviewers

With the Long No Exome LR dataset, we attempted to determine if long-molecule selection could also work on non-exome enriched cDNA. The Long No Exome LR dataset resulted in low sequencing throughput as described in the bar plot above, and thus **we chose to not include it in the manuscript, but added it in the response to reviewers with the hope that it may be informative.** We realize that this addition may have been confusing, and we apologize for not making the details about this additional dataset clear.

Secondly, we agree with the reviewer that 75-120 M reads are normal outputs for Oxford Nanopore Technology promethion flowcells. We also agree that our explanation of this observation was indeed a presumption, without sufficient evidence to make a strong claim for what caused it. The low number of reads could have been caused by a multitude of variables, including poor flow-cell quality. However, due to the observation of sequencing pores being clogged and flow-cell health deteriorating quickly (resulting in the low number of total reads), we felt long primer-dimers or other artifacts clogging the pores could be a potential cause, although again we lack evidence to confidently make this claim. This is why we chose not to add this data into the manuscript but added it to the previous response to the reviewer in hopes that it may be informative to the reviewer. **Overall, we cannot confidently claim what caused this observation and apologize for presuming without sufficient evidence.**

5B) Thanks for the information

We thank the reviewer for allowing us to make the manuscript better.

5C) They are still not really explaining how they are calculating “spliced (Fig 2b)” - presumably the exclusion set of intron-retaining. I really like Supp Fig 9, but then how does this dovetail with their bar plot just discussed? How are they getting roughly the same number of counts with and without capture, when they told me before capture improved their yield 10fold? I guess that Supp Fig 9 is downsampling the LR Exome data?

The reviewer asks several important questions and we apologize that they were not made clear. **We define a read as “spliced” if it contains at least one splice junction between exons.** In technical terms, a “spliced” read must be aligned to the genome in 2 or more alignment blocks separated by an intron. In other words, it must have at least one N section in the CIGAR string (skipped reference bases). Of note, minimap2 highly prioritizes introns with canonical dinucleotides at its ends and annotated splice sites when creating a spliced alignment.

Secondly, we point out that Supplemental Figure S13 (previously Supplemental Figure 9) compares the gene expression between **Standard LR** (naive long read) and **Standard Exome LR** (exome enrichment only) between a dataset from the same slide, which indeed have somewhat similar sequencing depths (Standard LR: ~100 million, Standard Exome LR: ~115 million). Thus, the average number of counts per gene is similar.

We also respectfully point out that the barplot the reviewer mentions consists of different data: **“Long Exome LR”** (exome enrichment + long molecule selection) and **“Long No Exome LR”** (**only long molecule selection without exome enrichment, which is not discussed in the manuscript** -- please see 5A for more information).

Although exome enrichment does increase the percentage of spliced reads (Figure 2b), overall gene counts and transcript counts remain similar (Supplemental Figure S13, new Supplemental Figure S12). **Overall, exome enrichment does not change gene expression, but rather increases the percentage of spliced reads from those genes.**

6) Instead of the plots included (2D, 2E) - I would like to have seen transcripts binned by their expected length plotted as 2D is. I am concerned that with the number of points and overplotting in 2D that we can't accurately perceive differences - you can't tell which in D are “long” vs “short” transcripts. It could - in principle - be that the long exome genes are getting 2 or 2.5 log counts while the standard exome is only getting 1 or so. Perhaps plotting the ratio - per transcript - of the counts for long exome over standard exome *versus* expected transcript length or observed alignment length might be a better way to attack this question?

We thank the reviewer for these great suggestions and have implemented it into 2 new supplemental figures, which suggest that **1. overall transcript expression is similar** between the Standard Exome and Long Exome datasets, and that **2. The ratio of Standard Exome LR/Long Exome LR transcript counts is not related to transcript length.**

1) We first examined the correlation of transcript expression between equally downsampled reads (n=2,000,000) from Standard Exome LR and Long Exome LR datasets (Supplemental Figure S12a). These two datasets were correlated, suggesting that overall transcript

expression is similar between the two (Pearson's $R = 0.69$, $p < 2.2e-16$). We have pasted this plot below for convenience.

2) We next examined the ratio of the reads per transcript of Standard Exome LR/Long Exome LR datasets and plotted this $\log_{10}(\text{ratio})$ by assigned transcript length (Supplemental Figure S12b). Transcript length was found using BioMart and includes UTRs and CDSs. **This plot revealed that the ratio of Standard Exome LR/Long Exome LR datasets is not strongly related to transcript length.** The median value for **Standard Exome LR/Long Exome LR ratio is 1.11 (where $\log_{10}(\text{Standard Exome LR/Long Exome LR ratio}) = 0.05$), suggesting highly similar transcript capture between the comparisons.** Also, there are 2,131 transcripts which are captured in the Standard Exome LR but not in Long Exome LR ratio (where $\log(\text{ratio}) = \text{inf}$), and 1,049 transcripts which are captured in the Long Exome LR but not Standard Exome LR (where $\log(\text{ratio}) = -\text{inf}$).

Overall, we have found that gene expression remains highly similar between the 2 methods (Figure 2D). We now also show that transcript expression between the 2 methods are correlated, and that transcript expression per method is not related to transcript length.

Supplemental Figure S12

Supplemental Figure S12: **a)** Correlation of transcript expression between equally downsampled Standard Exome LR ($n=2$) and Long Exome LR datasets ($n=2$). **b)** Log10 of ratio reads per transcript (Standard Exome LR/Long Exome LR) by assigned transcript length. Red dashed line indicates a value of 0.

7) Again - ok that % spliced doesn't change but what do the authors exactly mean by % spliced? Is this reads that have no introns at all, as defined by . what? Defined by GENCODE categories? Defined by the ORF? What? And if the ***point*** is that you have an increase in coverage across the transcript, then that

should be stated? But if you are filtering (as they say) to unique alignments, they are certainly discarding reads that are shorter, as those are likely multimappers? Could that explain the lack of difference between the samples - i.e. that they are bioinformatically filtering “short” molecules in the non-enriched datasets? I am guessing because there is a lack of clarity on what was done here to my reading.

We apologize for not being clear enough in previous answers. We respond to this comment in 2 parts: **1) Defining in detail how reads are processed, and 2) Responding to the question if we “are bioinformatically filtering ‘short’ molecules” by filtering reads, and conclude that the read filtering employed in this pipeline does not bias read length or splice sites.**

1. First, we define a read as “spliced” if it contains at least one splice junction between exons. In technical terms, a “spliced” read must be aligned to the genome in 2 or more alignment blocks separated by spliced junctions, i.e. have at least one N section in the CIGAR string (skipped reference bases). Of note, minimap2 highly prioritizes introns with canonical dinucleotides at its ends and annotated splice sites when creating a spliced alignment. We have now added this detail to the Methods section.

We now also add more details on how exactly the reads are processed:

Once barcodes are detected, the reads are mapped to the reference genome using minimap2 with the default setting used in IsoQuant (minimap2 -x splice --junc-bed annotation.bed -a --MD -k 14 -Y --secondary=yes). Spl-IsoQuant then assigns all mapped reads to the reference genes and isoform using the algorithm described in (Prjibelski et al., 2024). Multimapped reads are resolved by selecting the alignment that is the most consistent with the reference annotation or discarded. Reads that were uniquely assigned to a known gene are further processed to remove PCR duplicates.

2. The reviewer also asks an important question regarding if read filtering may artificially discard shorter reads. Here we point out that Spl-IsoQuant **does not discard secondary alignments and selects the best alignment with respect to the gene annotation**, which helps to include shorter reads in the analysis. Also, Spl-IsoQuant **only requires a read to be uniquely assigned to a gene**, which should not favor longer reads (compared to read-to-isoform assignment that certainly does). However, **a bias towards longer reads may be introduced during PCR deduplication**, where a read with the highest number of splice junctions is selected among reads with the same assigned gene, barcode and UMI sequences.

However, since only the UMI-filtered reads are used for further analysis (i.e., exon and transcript quantification, differential-splicing analysis) **it is important to demonstrate what effect molecule enrichment has on the final usable set of UMI-filtered reads.** Reads that are unmapped, not assigned to a gene, do not have a barcode or are filtered during UMI-filtering (PCR deduplication) are unusable and should not be included in the general statistics. Answering the reviewer’s question, we also present the average read length and % spliced

(the percentage of reads with at least one bona-fide removed intron) for raw reads and UMI-filtered reads from different experiments (Standard long-read sequencing, Standard exome and Long exome) side by side in a new Supplemental Figure S17. Overall, these properties look fairly similar for both raw and UMI-filtered reads, thus concluding that **read filtering implemented in the pipeline does not introduce any significant bias with respect to read length or splice sites.**

Supplemental Figure S17

Supplemental Figure S17. **a)** Average read length per sample separated by all reads (raw) and UMI filtered. **b)** % of spliced reads per sample separated by all reads (raw) and UMI filtered.

7B) Further, I would like to see the distance of these reads to the expected TSS, or the aligned read length compared to the “reference” transcript length. Presumably that, and the coverage across transcript, improves with capture? If that is the point, it should be made more clear through other data viz.

We thank the reviewer for this valuable suggestion. As recommended, we have added a new Supplemental Figure S15, which describes transcript coverage fractions for 2 samples sequenced with 3 different methods (Standard LR, Standard Exome LR and Long Exome LR) below. **These plots show that Long Exome LR sequencing substantially improves on the fraction of captured transcripts** as it has a noticeably higher percentage of reads covering at least 50% of the reference transcript (Supplemental Figure S15a-b). For example, Long Exome has roughly 3 times more reads (percentage-wise) that capture the entire transcript

compared to the other 2 methods. To verify whether the same observations hold for spliced transcripts, **we provide the same plots calculated only using spliced transcripts** (here “spliced transcript” is defined by the gene annotation). As these plots show, the difference between these 3 sequencing methods becomes even more pronounced (Supplemental Figure S15c-d).

Furthermore, we investigated how average transcript coverage changes across various transcript lengths. As the plot below shows, Long Exome LR experiments allow us to capture larger portions of transcripts. The difference becomes more distinguished for longer molecules (Supplemental Figure S15e-f). The same observations hold for spliced transcripts with the difference between Long exome and the other 2 methods being even slightly more visible (Supplemental Figure S15g-h). Once again, we thank the reviewer for these insightful suggestions and have added the above plots to Supplemental Figure S15.

Supplemental Figure S15

Supplemental Figure S15. a) % of reads which fall into bins of fraction of transcript covered in sample 1 across standard, standard exome, and long exome datasets. **b)** Same as (a) but for sample 2 across standard, standard exome, and long exome datasets. **c)** % of spliced reads which fall into bins of fraction of transcript covered in sample 1 across standard, standard exome, and long exome datasets. **d)** Same as (c) but for sample 2 across standard, standard exome, and long exome datasets. **e)** Average fraction of transcript covered binned by reference transcript length covered in sample 1 across standard, standard exome, and long exome datasets. **f)** Same as (e) but for sample 2 across standard, standard exome, and long exome datasets. **g)** Fraction of average spliced transcript covered binned by transcript length covered in sample 1 across standard, standard exome, and long exome datasets. **h)** Same as (g)

but for sample 2 across standard, standard exome, and long exome datasets. Sample 1 indicates sample 455 and sample 2 indicates sample 5391.

8A) Sorry - are the authors in Supp Table 1 simulating various coverage of full-length transcripts using truncation rate, or taking the possible transcripts, simulating truncation percentages, then applying size cutoffs to simulate what they did in their *experiment*. The procedure used to generate these simulated reads is not fully clear to me from the description. In my opinion, they should be simulating a population of transcripts with different truncation fractions, then applying size cutoffs to the molecules, not simulating different truncation percentages alone (which it appears they did from the description.

Yes, in Supplemental Table 1 generated from the previous response to reviewers, we only simulated different truncation percentages without simulating any enrichment or size selection. The goal of this truncation simulation was to address the following question from the previous review: “Basically, how long do reads need to be to get accurate isoform quantification?” and estimate the relation between quantification accuracy and read truncation. **As the reviewer suggested, to mimic real-life experiments we performed another set of simulations** (see below).

Since none of our sequencing methods has an explicit molecule size cutoff, to mimic spliced molecule enrichment and longer molecule selection we simulated ONT data following empirical (1) read length distribution and (2) transcript abundances obtained from real experiments:

1. Standard: usual long read sequencing without molecule selection;
2. Standard Exome: spliced molecules regardless of their size;
3. Long Exome: spliced molecule with longer reads being preferred.

Assignment precision and recall are presented in the table below. While selection for spliced reads clearly affects recall in a negative way (isoforms with multiple exons tend to be more

Sequencing method	Average length, bp	Isoform assignment recall, %	Isoform assignment precision, %
Standard	516.6	55.9	90.6
Standard Exome	446.9	30.4	77.5
Long Exome	956.9	42.5	92.0

challenging to assign), while selection of longer molecules yields higher precision and recall.

9A) Is this simply due to read depth? What does the saturation curve look like? What's the reads/UMI

The reviewer asks an important question regarding if the previously added results (Figure 2F, Supplemental Figure S22) could be driven by sequencing depth. In order to answer this question, we have generated a new Supplemental Figure S4, which shows saturation curves of the Long Exome as well as the Naive SR data. This figure shows that the unique gene-barcode pairs (meaning the total number of unique genes detected per barcode, summed across all barcodes), in the short-read data is more saturated than the long-read data, as expected. However, the long exome data has more intron barcode pairs, as we demonstrated in Figure 2F. On average, the Long Exome data contains 46.93 reads/gene-barcode pair and the Naive Short Read data contains 57.5 reads/gene-barcode pair.

Supplemental Figure S4

Supplemental Figure S4. a) Unique gene-barcode pairs in Naive SR data plotted in bins per fraction of data processed. b) Unique gene-barcode pairs in Long Exome LR data plotted in bins per fraction of data processed.

9B) How is it possible that these numbers for UMIs/barcode are so similar, given that the number of reads (S7) looks so different? Perhaps collecting all this information in a table would aid in understanding what the authors are looking at - reads per sample, UMIs/read, UMIs/barcode &/or reads/barcode, etc.

We thank the reviewer for this question and implement their suggestions below. First, to answer the question “How is it possible that these numbers for UMIs/barcode are so similar,

given that the number of reads looks so different” the reviewer is correct that there is substantial variation between the overall number of spliced reads between datasets. However, this number becomes overall more similar once only mapped, barcoded, spliced, and UMI corrected reads are selected and examined. Thus, overall read numbers can vary due to ONT variables such as flow-cell health. However the Spl-Isoquant read processing of identifying uniquely mapped, spliced, and barcoded reads paired with deduplicating UMIs removes uninformative reads and overall detects roughly similar proportions of UMIs/barcode. As the reviewer has suggested, we’ve included a **new Supplemental Table 2** indicating all of these values per sample below. Note that column headers “UMI” indicate UMI corrected reads and “BC” indicate barcoded reads.

Supplemental Table 2

Sam ple	Method	Total reads	Uniquely assigned + spliced	Uniquely assigned + spliced + BC	Uniquely assigned + spliced + BC + UMI	UMIs (spliced + BC +UMI) /BC	UMIs (spliced + BC +UMI) /gene	Avg. Exons per UMI (spliced + BC +UMI)
455	Standard	41,972,986	8,433,910	3,856,235	1,077,750	21.53	67.67	3.53
455	Standard exome	122,159,255	76,615,709	23,086,992	3,724,059	54.09	228.72	3.56
455	Long exome	84,786,198	65,716,027	28,571,114	1,107,848	21.29	80.34	5.89
5391	Standard	100,138,443	31,171,014	5,864,339	1,711,503	27.57	95.12	3.62
5391	Standard exome	115,888,900	79,132,036	21,078,824	3,368,247	47.76	185.10	3.76
5391	Long exome	81,702,222	46,978,335	21,989,777	2,700,085	49.58	147.72	4.13

9C) I remain surprised that they *need* that many molecules for the clustering? Wouldn’t a smaller set be sufficient in many cases? Did the authors compare clustering between short and long read datasets? You showed and argued convincingly that there is no bias in quantitation between short and long data \- so why is short needed for clustering beyond having “more”.

The reviewer brings up an important point which we apologize for not addressing fully. As we previously showed, there is no bias in quantification between Naive Short Read and Long Exome LR (Supplemental Figure S8). Indeed, there was an overall high correlation between

the SR and LR, but the overall counts in LR were lower, as expected. Although for cell-type decomposition programs more data is generally better in order to yield the highest accuracy possible, the reviewer is correct that we did not attempt to define cell types with the long-read data from the beginning as the short-read data provided a more in-depth gene expression dataset. **Here, we show that attempting to use a Long Exome LR dataset as input into the cell-type deconvolution program severely limits the number of cells identified due to decreased number of UMIs per barcode** (Supplemental Figure S5). Using the long-read data as input only results in identifying 17% of the originally identified cells (Supplemental Figure S5e), and only identifies 1,098 excitatory neurons as compared to the previous 20,787 (Supplemental Figure S5f). We hope the reviewer agrees that using the long-read data as input for the cell-type deconvolution program would not have been sufficient in order to identify cell types and individual cells.

Supplemental Figure S5

Supplemental Figure S5: **a)** UMIs per spot from Naive short-read data plotted by spatial location. **b)** UMIs per spot from Long Exome data plotted by spatial location. **c)** Excitatory-neuron singlets and their positions identified from inputting Naive Short Read data into a deconvolution program. **d)** Excitatory-neuron singlets and their positions identified from inputting Long Exome data into a deconvolution program. **e)** Number of identified “Doublet Certain”, “Doublet Uncertain”, “Reject”, or “Singlet” spots by data input. **f)** Number of identified excitatory-neuron and oligodendrocyte singlets by data input.

10\) I still feel like this is nice, but it adds to the papers dual focus - the actual biology (we are about to enter that part, and the methodological work.

We thank the reviewer for allowing us to clarify this issue.

11A) I get the point about modifying a third-party tool, but I also feel like there were lessons to be learned from wf-single-cell here, since it just is a nf-flow workflow. For example, they identify the parts of the adaptor using parasail which implements (IIRC) a Needleman-Wunsch alignment to find the BC and UMI chunks. Instead the authors *apparently* reinvented parasail? My concern is primarily for reproducibility of the work, but the authors have an established track record of well-supported tools, so I guess justifying why such a tool is needed is the biggest key.

We apologize for not being clear enough. We use the exact k-mer matching method only to quickly detect a short list of potential barcode candidates and further use the Smith-Waterman algorithm to detect the best candidate. Since the k-mer length is much shorter than the barcode length, **we thus account for possible errors and truncations**. As for the implementation, instead of parasail we used another library called ssw-py (Zhao et al. 2013), which effectively implements the Striped Smith-Waterman algorithm in C using SIMD instruction (similar to the parasail implementation) and wraps it up as a Python library. We have now clarified these details in the Methods section as well.

For reproducibility, we published our pipeline on GitHub and intend to continue supporting the developed software along with the IsoQuant package. Although, currently, the entire package is implemented in Python (due to IsoQuant heritage), we plan to create a NextFlow pipeline in the future.

The main reasons for developing this software on the basis of IsoQuant are: (1) there is no software that supports barcode calling for long-reads generated with the Curio protocol; (2) modifying existing tools created by other researchers may be time-consuming and (3) it allows us to tune the algorithm to achieve the desired precision/recall trade-offs and output necessary information.

11B) However, I would add (based on my reading of their methods) they are demanding exact matches, which is likely to lose reads that have errors. The sockeye/wf-single-cell approach generates a shortlist of high-quality barcodes then collapses based on error and likelihood of the reads to those. I suggest that's a more powerful approach than described here.

Again, we apologize for the lack of clarity — our algorithm does allow inexact matches, the details are given above in the response to 11A and also clarified in the Methods section.

Regarding the high-quality shortlist of barcodes and two-step barcode detection. In 10x protocols, the number of barcodes in a general whitelist (~700'000) dramatically exceeds the number of actual cells sequenced (~5'000-10'000). Therefore, a 2-step barcode-detection algorithm significantly shrinks the search space. However, in this project, Curio already provided us with shortlists of high-quality barcodes for each sequencing experiment (often referred to as "whitelists" in the manuscript). The number of barcodes in these whitelists (~70'000 on average across multiple experiments) are very close to the number of actual spots sequences by long reads (~60'000 on average). **Thus, pre-selecting an additional shortlist of high-quality barcodes will not shrink the search space and therefore will not have a major effect on the sensitivity of the barcode detection procedure.**

12\| Why are the authors arguing for a 4.4% error rate? Latest basecallers (I am *not* suggesting they need to recall their data) have a much improved error rate - in discussion points they should point out that their (and other) algorithms are likely to improve recall with that improved error rate. But I still argue (as I said above in 11B) that they should generate a white list and perform barcode correction rather than a single round.

We came to the 4.4% using an error rate estimation via read alignment to the reference transcriptome. In this work we used Oxford Nanopore Technology's Dorado basecaller v0.9 (basecalling model: dna_r10.4.1_e8.2_400bps_sup@v4.2.0), which was the latest available version at the time of sequencing.

The error rate was estimated in the process of training the Trans-NanoSim (Hafezqorani et al., 2020) model and further used for simulation. This software computes the error rate by mapping long reads directly to the transcriptome and then counting mismatch, deletion and insertion errors. Although such a simplistic method does not account for variants and RNA editing, it is unlikely to have dramatic effect on error rate estimations. Also, based on our experience with ONT data, even the latest datasets feature at least a 3% error rate when measured via direct alignment.

We have now clarified the reason behind a single-round barcode detection in response to 11B.

13\| Yes, this makes a lot of sense, and about what I expected. Thanks for adding this to the Methods.

Thank you for letting us expand on this description.

14\| I *do* agree that it's fine to use a short read to generate layers. But I think it's instructive to *show* not *tell* the reader (and me) how it works to define layers with long-read alone. It can be a supplemental figure, but you should

explain in the manuscript this point, that you were focusing on splicing with long read and layers with short.

We thank the reviewer for this comment and apologize that we did not make this point more clear previously. We agree that we should have included this information as a supplemental figure. **We have implemented layer-specific marker expression across a slide in both Naive Short Read and Long Exome LR data into a new Supplemental Figure S3** (also described in answer 1A). Additionally, we also used other information to define layers, such as excitatory neurons and UMI density. In order to see if the Long Exome LR data had sufficient amounts of data to describe these layer changes similarly to the Naive Short Read data, we performed cell-type deconvolution again using the Long Exome LR data as input. We found that this analysis **severely limits the number of cells identified due to the decreased number of UMIs per barcode** (Supplemental Figure S5, see answer 9C). We hope the reviewer agrees that using the long-read data as input for the cell-type deconvolution program would not have been sufficient in order to identify cell types.

Supplemental Figure S3

Supplemental Figure S3: a) Log10(UMI) counts of Naive Short Read data per spot of gene CALB1 plotted by spatial location. **b)** Log10(UMI) counts of Long Exome data per spot of gene CALB1 plotted by spatial location. **c)** Log10(UMI) counts of Naive Short Read data per spot of gene RORB plotted by spatial location. **d)** Log10(UMI) counts of Long Exome data per spot of gene RORB plotted by spatial location. **e)** Log10(UMI) counts of Naive Short Read data per spot of gene MBP plotted by spatial location. **f)** Log10(UMI) counts of Long Exome data per spot of gene MBP plotted by spatial location. Black lines indicate layer cutoffs.

Supplemental Figure S5

Supplemental Figure S5: **a)** UMIs per spot from Naive short-read data plotted by spatial location. **b)** UMIs per spot from Long Exome data plotted by spatial location. **c)** Excitatory-neuron singlets and their positions identified from inputting Naive Short Read data into a deconvolution program. **d)** Excitatory-neuron singlets and their positions identified from inputting Long Exome data into a deconvolution program. **e)** Number of identified “Doublet Certain”, “Doublet Uncertain”, “Reject”, or “Singlet” spots by data input. **f)** Number of identified excitatory-neuron and oligodendrocyte singlets by data input.

15) Ok, you are saying it won’t be as accurate, but could you try in a single sample or a subset of locations to just demonstrate it? Or at the least say it directly in the manuscript?

We have incorporated the reviewer’s suggestion of using long-read data to deconvolve cell types into supplemental figure S5, as answered above and in 9C. We thank the reviewer for their suggestion which has strengthened the manuscript.

16\) I appreciate this point, but the authors need to be more clear in the manuscript about the weakness of the Curio platform for long-read data given these limitations (the non-TSO approach).

The reviewer is correct that the previous version of the manuscript lacked a sufficient discussion of the methodological limitations, and we thank them for pointing this out to us. We have since added to the discussion section about the lack of full-length molecules and how future approaches could modify this and enhance the field.

“ Additionally, while this technology increases the average length and transcript coverage per read compared to the standard, most reads are not full-length. This drawback limits the extent to which we can describe 5' TSSs and full-length isoform changes across development. Other technologies, including but not limited to 10X Genomic's Visium, DBiT-Seq, and StereoSeq, use a transcript switch oligo (TSO) rather than a universal priming site (UPS) during 2nd strand synthesis which would enable full-length molecule capture. However, other variables including spatial resolution (spot size), molecule diffusion, and capture efficiency should also be considered when choosing a spatial transcriptomic technology. Future work should focus on optimizing high-resolution and low diffusion spatial technology for mapping not only of full-length isoforms across a region, but also other modalities at the same time. This would propel the field forward and enable the investigation of multi-modal networks in relation to their spatial location and microenvironments.”

17\) Great

Thank you!

18\) Since you removed it I've moved on, but I think using unique assignment, as I think I've previously said, is a mistake in part. I direct the authors attention to oarfish and kallisto-lr \- kallisto-lr I believe even uses exom capture as part of their assignment.

We thank the reviewer for the insight — we have now changed the strategy so that the read just needs to be assigned to a gene to be considered for further analysis. This, indeed, makes more sense.

Referee #1 (Remarks on code availability):

Code runs

REVIEWER 2

Referee #2 (Remarks to the Author):

In the revised manuscript, the authors address my technical comments. For my comment on biological significance, the authors elaborated on the biological relevance of their results and also presented some additional biological results, but the significance of these results is still unclear. Ultimately, it is the editor's decision whether the results are of sufficient significance for publication in Nature.

We thank the reviewer for their comments which have improved the quality and thoroughness of this manuscript, including now describing a lack of bias in transcripts measured with our method as well as clarifying key text to make the work more approachable to the biology community as a whole. We understand the reviewer's concern that the biological significance of the findings remained unclear even after the addition of new work, and we agree that these findings were indeed interesting, however disjointed in forming a complete mechanistic story. We hope that the reviewer agrees that this manuscript is sufficient for acceptance to Nature Communications.

REVIEWER 3

Referee #3 (Remarks to the Author):

The authors have addressed all my main comments. In particular, the authors have performed additional analyses to show that enrichment for longer cDNAs during sequencing library preparation for spatial isoform sequencing (Spl-ISO-seq) does not lead to a systematic bias in the detection of transcript levels. The new data are shown in Fig. 2D, 2E and Supplementary Fig. S10. The authors now also clarify the limitations of the Spl-ISO-seq approach in the main text, which will be helpful for future users of the method. All the additional clarifications in the main text have further improved the manuscript and will also allow researchers outside the neuroscience community to assess the key biological findings.

I think the new approach, including a software tool for analysing Spl-ISO-seq data, and the interesting new insights into the heterogeneity of RNA processing between different cell layers of the human cortex and developmental stages, are of general relevance to the RNA processing and neuroscience communities, as well as to the growing field of single-cell omics. The new approach can now be applied to other complex human tissues to investigate the heterogeneity of alternative splicing and polyadenylation site usage in different cell layers, and how this may change in disease states. This approach is also likely to inspire new protocol variants, eventually leading to techniques to study the variability of RNA processing between individual cells in tissues, even within the same cell layer.

We thank the reviewer for their comments which have improved the quality and thoroughness of this manuscript, including now describing the lack of bias in transcripts measured with our method as well as clarifying key text to make the work more approachable to the biology community as a whole. We also thank the reviewer for recognizing the various biological questions and applications that this method could be used for in the future. We hope that the reviewer agrees that this manuscript is sufficient for acceptance to Nature Communications.

REVIEWER 4

Referee #4 (Remarks to the Author):

We appreciate the authors' efforts in responding to our comments and addressing some of the concerns, particularly their comparison of the short- and long-read approaches for detecting splicing. We fully agree that short-read sequencing is inadequate for meaningful splicing analyses in spatially resolved single-cell contexts. However, we are not convinced that Spl-ISO-Seq solve these issues. Basically, if, as the authors clarified, most cDNA molecules are not full-length, then this method will not reliably identify alternative splicing isoforms, as it only captures a biased subset of isoforms. Related to that, we still find the manuscript lacking some robustness checks regarding alternative splicing calculations. Moreover, we still do not see how the "joint usage of [two approaches]" already used (and developed) justifies the claim of developing a novel technology. A title emphasizing splicing may be more appropriate, but the claim of a new technology remains unconvincing. In addition, regarding the addition of mechanistic insights and physiological relevance, we do not see any improvements in how these analyses enhance our understanding of brain changes during puberty. While we appreciate the mechanistic insight provided by the repetitive elements, it does not add to the study's overall relevance. Moreover, the association between alternative splicing and alternative polyadenylation has been extensively documented before, and without any new physiological relevance, this remains a limited addition. Overall, we believe that without achieving full-length sequencing, this methodology may introduce significant biases.

General Response to Reviewer #4:

Dear Reviewer 4,

We are very grateful for the time and effort which you put into reading over our manuscript a second time. Your questions and comments have allowed us to clarify and expand on a multitude of topics, gain stronger insights, and modify the manuscript to, hopefully, a more clear and informative form for the future readers. These efforts have resulted in a total of 7 new supplementary figures, 1 new supplementary table, and substantial additions to the text's

discussion. Overall, we hope the reviewer agrees that this manuscript is now sufficient for acceptance to Nature Communications.

In summary, we were able to (i) emphasize the necessity for both short- and long-read sequencing in several new supplemental figures as well as the discussion, and (ii) conduct simulations to assess false positives and negatives.

Below we provide point-by-point answers to all comments. Reviewers' questions are written in normal font, the answers are written using *dark blue font* with the new parts of the manuscript highlighted with *cursive*. We also highlight the key summary points in **bold**.

Specific comments:

1. Since, even after introducing long-read sequencing, most cDNA molecules are not full-length, and the analysis only includes those with an annotated TSS at the very 5' end (which is not explicitly stated by the authors but inferred from their comments), the alternative splicing isoforms considered are inherently biased. This bias arises because the method predominantly captures isoforms with internal TSSs that are less prone to fragmentation.

We thank the reviewer for this comment and apologize that we were not clear enough in previous answers. We respond to the comment “the analysis only includes those with an annotated TSS” in detail below. Overall, we point out that molecules **do not need to have an annotated TSS** in order to be included in this manuscript’s major analyses. Indeed, the major statements made in this manuscript are supported by analyses that work on an exon, rather than isoform, basis.

We respond to the reviewer’s comment “the analysis only includes those with an annotated TSS”. **We respectfully point out that molecules do not need to have an annotated TSS in order to be included in this manuscript’s major analyses, which are supported by statistical tests for individual exons and poly(A) sites (not complete isoforms).** We apologize that this was not made clear and hope to clarify this point now. In this manuscript, most exon-based analyses include all reads, which can be assigned to a gene, spliced, barcoded, and UMI corrected. Only in the full-length isoform analysis (Supplemental Figure S26) were exclusively full-length transcripts including TSS considered.

The reason that most cDNA is not full-length is due to Curio Bioscience’s established library preparation protocol, where a 5' random hexamer (termed UPS) binds to a random portion of the gene rather than a 5' template switch oligo (TSO). **Thus, most molecules (> 95%) are not full-length as the original library preparation protocol was optimized for short read sequencing and gene expression analysis only and was not designed to capture full-length molecules.** Please see below for the protocol overview diagram which illustrates how

cDNA is generated. We note that this image is from the Curio pipeline and not in the manuscript.

2. In the short-read data, is fragmentation performed after barcoding? If so, wouldn't much of the cDNA derived from gene bodies be lost? Is there a specific reason to why not modify short-read approaches to retain more gene body reads? Further justification would be helpful for the claim that Spl-ISO-Seq is uniquely required, rather than other strategies that enrich for long, spliced molecules.

The reviewer is correct in their assumption and poses an interesting question regarding cDNA synthesis and short-read library preparation. Here we go over multiple answers to the questions "is fragmentation performed after barcoding?" and "Is there a specific reason to why not modify short-read approaches to retain more gene body reads".

- 1. First, in response to "is fragmentation performed after barcoding?": Yes, fragmentation occurs after the cDNA amplification (see diagram in answer #1) as a step within the Illumina Short Read library preparation. The cDNA which is generated from the cDNA amplification step is split into 2 pools, 1 for short-read library**

preparation (including fragmentation) and sequencing, and one for exome enrichment, long-molecule selection, and long-read sequencing.

2. **Secondly, in response to, “If so, wouldn’t much of the cDNA derived from gene bodies be lost?”: Yes, the reviewer is correct in this assumption.** Through fragmentation, portions of the gene bodies are lost, as is a common drawback to short-read sequencing. This is a major reason why we perform long-read sequencing, as we do not fragment those molecules and capture much more information per molecule.
3. **Lastly, we answer the question “Is there a specific reason to why not modify short-read approaches to retain more gene body reads” below.** Although short-read sequencing could be modified, here we go through why these would not be optimal for this method and why long read sequencing is required instead. These potential modifications include **(i)** fragmenting molecules before barcoding rather than after — which could negatively affect RNA quality and only fragments containing Poly(A)s would be barcoded, **(ii)** increase the fragment length of short-read molecules by reducing or eliminating fragmentation all together -- which would reduce the amount of near-Poly(A) sequences captured and require read length normalization. Additionally, the sequencing settings for Illumina would continue to limit the number of basepairs sequenced per read, regardless of the read length (maximum of 300 bp per read for the machine used here). **We additionally thank the reviewer for allowing us to justify this method in more detail, which we add to the discussion.**

Below we go over these reasons in more detail:

i. **First, if we fragmented molecules before barcoding rather than after, this would not be possible using this technology for several reasons.** Importantly, we note that **this would fragment the RNA itself** as this would occur before the reverse transcription and 2nd strand synthesis steps in the protocol. Primarily, this would not be feasible due to the fact that RNA in fresh-frozen, non-fixed, human tissue degrades quickly, especially once it has been sliced to 10um sections and placed on the Curio Seeker barcoded slide at room temperature. **Adding the extra step of RNA fragmentation early on would increase RNA degradation and decrease sample quality overall, making the analysis of long molecules (either with long or short read sequencing) more challenging.** Additionally, in the Curio Seeker protocol, barcodes hybridize to the RNA molecules’ Poly(A) tails (see diagram in Answer #1). **Thus, even if molecules were fragmented, only the fragments which contain the Poly(A) tail would be captured and barcoded,** which would highly limit the amount of information captured per read.

ii. Additionally, we could perform short-read sequencing by reducing or eliminating fragmentation all together. By extending the fragment length, we would no longer sequence near-poly(A) sequences, and further require fragment lengths to be normalized. We also note that simply extending the length of the fragment will not affect the number of bases which will be sequenced as **the maximum length per paired end for the Illumina NovaSeq X** (the short-read sequencing machine which was used) **is 150 bases per paired end** (so 300 bases maximum per fragment). **Thus, even though the molecule itself would be longer, the**

length of the sequenced read would not change, and thus not add more information regarding the gene body.

Perhaps if a different sequencing machine and/or short-read library kit were used, longer paired-end sequencing settings may be possible to attain. However, a high proportion of these reads may be intronic, which would remain uninformative for splicing purposes despite the increase in sequencing length. A way to avoid intronic reads would be to perform an exome-enrichment, similarly as we did to the long-read data. However, performing an exome-enrichment on short read data could slightly bias the “ground truth” of gene expression per cell which we require for cell-type and layer assignments. Additionally, this would increase the cost per read dramatically, especially at the read depth which is necessary for gene expression analyses. Long molecules are also unnecessary for the purpose of measuring gene expression, which can often be mapped with only 50-100bp fragments. **Overall, long-read sequencing is the most time and money effective way to capture entire spliced molecules, while short-read sequencing is optimized for identifying genes by sequencing short fragments at a high read depth.**

However, we agree with the reviewer that many interesting biological variables can only be measured by capturing an entire read, including alternative exon inclusion. **This is one of the major reasons why we do long read sequencing.**

We apologize that this reasoning was not made clear to the reviewer previously, and have added more about this topic to the discussion:

“Of note, cell-type information was determined with short-read sequencing of fragmented cDNAs due to enhanced read depth and gene expression profiles using short reads. Long-read sequencing (here of unfragmented cDNAs), although necessary to describe transcript information spanning multiple exons, generally is not sufficient for gene-expression based analyses (unless multiple flow cells are used to match short-read depth per sample). Vice versa, short-read sequencing is not sufficient to identify alternative exon inclusion nor characterize fully synthesized molecules. Thus, both sequencing modalities are required in tandem for Spl-ISO-seq.”

3. What is meant by a “known intron-barcode pair”? Does this refer to a barcode-containing read spanning an annotated exon-exon junction?

We apologize for not being clear enough previously. A “known intron-barcode pair” corresponds to a detected splice junction that has previously (among already analyzed reads) been detected with the same barcode. Thus, the y-axis in figure 2F sums the total number of splicing events per barcode. We have edited the text to make this more clear to the reader:

“We then explored whether short-read analysis alone could serve for splicing analysis in our brain slices. To this end, we counted the number of detected splicing events per barcode and added these from all barcodes, which we termed intron-barcode pairs.”

We hope we have clarified this for the reviewer.

4. The exclusion threshold of exons with total reads < 10 seems too low. The authors should first explore different read filtering thresholds to assess the impact on results, conduct comparisons with simulations, and measure false positive and false negative rates. While this is crucial for any method aiming to measure alternative splicing, it is especially important when proposing a “new technology” that integrates short- and long-read data in spatial transcriptomics.

We thank the reviewer for this thoughtful question and allowing us to investigate further into the statistical sensitivity of this method. We agree that the exclusion threshold of exons <10 reads could be low, especially for exons which have a low dPSI. We hope the reviewer agrees that we thoroughly investigate this question in 3 parts. I) First, **we perform simulations and find that 10-19 reads per comparison are required to detect any splicing changes, and those that are detected have $\text{abs}(\text{deltaPSI}) \geq 0.3$.** II) Next, we investigate patterns in our own data and find that a **minimum of 21 reads total are found to detect splicing changes**, despite the 10 read minimum cutoff. III) Lastly, **we re-perform exon tests with a larger cutoff** and investigate differences between the datasets, and conclude that increasing the cutoff slightly reduces the number of exons which are originally measured, but does not change the exon set that passes the criteria to be tested (chi-squared) and significant. **Thus, increasing this cutoff does not change significant exons that we detect.**

I) First, we performed simulations to investigate if statistical significance (corrected for multiple comparisons) could be reached with <10 reads per condition and a large deltaPSI.

In order to investigate this, we simulated a large number of 2x2 matrices containing exon inclusion and exclusion counts with a deltaPSI of 0.1 (1000 total counts in each column). All such matrices have p-values $\leq 10e-5$. We then downsampled these to combined counts of 0-9, or 10-19, or 20-29, 30-39, 40-49, and 50-249 per comparison, and recorded the fraction of matrices that pass Benjamini-Yekutieli correction for multiple testing (at corrected FDR < 0.05 for 100 tests in each case). These fractions give an idea of how many reads are required to find a truly existing deltaPSI of 0.1.

We repeated this process for deltaPSI 0.2, 0.3, 0.4, 0.5, and 0 (to assess false positives). In summary, for deltaPSI ≥ 0.4 and read numbers (in each column) ≥ 30 , one reaches a sensitivity of 79%. Additionally, these simulated results demonstrate that a **deltaPSI of 0.5 requires a minimum of 10-19 reads per condition to be detectable.** We have added this table as a new Supplemental Figure S23a and paste it below for convenience.

II) Next, we wanted to confirm these simulated results with the observations in our own data as well. In the significant exons from the Cortex Child vs. Young Adult exon inclusion tests, we plotted the detected deltaPSI by the total number of reads per significant exon, which we added to a new Supplemental Figure S23b. We find that the **minimum number of total reads**

in order for an exon to reach an FDR <0.05 is 21, in which the abs(dPSI) detected was 1. This observation aligns with the simulated results.

III) Lastly, we wanted to examine if increasing the threshold of reads required would dramatically change the number of exons tested and the % of significant exons. We wanted to increase the threshold to remove exons that should not be tested due to too few reads, but also not increase the threshold so much that significant exons would be filtered out. Thus, we chose to increase the read threshold to 20, rather than the previous 10 for the Layer 4 Child vs. Young Adult comparison. Whereas with the original threshold 28,544 exons with ≥ 10 reads were detected, 8295 passed the chi-squared criterion for testing and 1918 with $FDR < 0.05$, now 16594 exons with ≥ 20 reads were detected, the same 8295 passed the chi-squared criterion for testing and the same 1918 with $FDR < 0.05$. **Thus, increasing the number of reads threshold to 20 did not affect the results.**

Supplemental Figure S23

a

Number of Reads Per Condition (Child vs. Young Adult)	IdPSII					
	0	0.1	.2	.3	.4	.5
50-249	0	18	63	87	98	100
40-49	0	0	12	65	90	99
30-39	0	0	1	34	79	90
20-29	0	0	1	14	43	78
10-19	0	0	0	3	14	38
0-9	0	0	0	0	0	0

b

Supplemental Figure S23: **a)** Matrices with 1000 reads per condition with predefined $|\delta\text{PSI}|$ are downsampled to read counts of [0,9], [10,19], [20,29], [30,39], [40,49], [50,249] per condition. Values indicate the fraction of matrices that pass Benjamini-Yekutieli correction for multiple testing per $|\delta\text{PSI}|$ and read number combinations. **b)** Significant exons from the Cortex Child vs. Young Adult comparison plotted by $|\delta\text{PSI}|$ and $\text{Log}_{10}(\text{Total Reads})$.

5. While the authors highlight a 55-fold improvement, additional comparative analyses would provide more insight into the nature of this improvement. For example, a meta-gene plot illustrating the coverage of reads with splicing information in short- vs. long-read approaches could clarify whether the improvement stems from increased depth, coverage, or both, as well as any specific limitations in detecting splicing at the 5' end of genes.

The reviewer offers a great suggestion which we have now implemented as a new supplemental figure. We plotted the normalized read coverage across all annotated genes for the Naive Short Read, spliced Naive Short Read (subset of Naive Short Read which is spliced), and Long Exome LR (exome enriched + long molecule selected long read) data (Supplemental Figure S16) in one sample. This plot demonstrates that although the Naive Short Read has high coverage in the transcription end site (TES, equivalent to Poly(A) site) especially, coverage is not uniform and dramatically decreases to the transcription start site (TSS). In the Long Exome Long Read data, coverage at the TES is lower compared to the short read data, as expected, however coverage along the continuation of genes is more uniform and slowly drops off. We also examined what the coverage looked like in the subset of spliced reads from the Spliced Naive Short Read data. Here, overall coverage is extremely low, with a slight peak at the TES, although much smaller than the other 2 datasets. Overall, Long Exome LR shows the most consistent coverage of genes, Naive Short Read the highest depth, although coverage is mostly at the near TES, and the Spliced Short read lacks both coverage and depth.

Supplemental Figure S16. Metagene plot showing exonic coverage of annotated genes with normalized length. Red shows coverage from the Naive Short Read data, blue trace shows coverage from the Long Exome LR data, and green trace shows coverage from the Spliced Short read data. TSS indicates transcription start site and TES indicates transcription end site.

6. Regarding the connection between AS and APA, how does defining regulated APA sites as having a $|\text{dPI}| > 0.2$ and regulated alternative exons as having a $|\text{dPSI}| > 0.2$ ensure that the analysis is not biased toward highly expressed genes?

We thank the reviewer for asking this important question and apologize for not making this more clear in the text. By setting a $|\text{dPI}|$ and/or a $|\text{dPSI}|$ cutoff rather than a significance cutoff where $\text{FDR} < 0.05$, we are examining changes which 1. are **substantial, however may lack sufficient power to reach significance**, while 2. Also **filtering out false positives** which reach significance only due to read number, but lack sufficiently large $|\text{deltaPIs}|$. For example, if a gene has 30 total reads (15 per each comparison) with a $|\text{deltaPI}|$ between 0.2 and 0.3, it will likely not reach significance, as suggested by our simulations performed in question 4. Additionally, genes with extremely high expression may reach significance without a substantial change in dPI or dPSI , simply due to an increased number of reads. If a gene has 2000 total reads (1000 per each comparison) with a $|\text{deltaPI}|$ of 0.05, it is highly likely to reach

significance and should be considered as a false positive. We have added more explanation to the results section in order to clarify this for the readers:

“We separated genes into 2 groups based on those which lack developmentally regulated Poly(A)-sites ($|\text{deltaPI}| < 0.2$, “non-changing”) and those which undergo developmentally regulated Poly(A)-sites ($|\text{deltaPI}| > 0.2$, “changing”). We defined these groups based on $|\text{deltaPI}|$ criteria rather than significance in order to limit the number of false positives while also capturing substantial changes which may lack sufficient power to reach significance.”

Overall, by setting a $|\text{dPI}| > 0.2$ cutoff, we are examining changes which are substantial, although perhaps lowly expressed, while also limiting false positives.